# B-SOiD, an open-source unsupervised algorithm for identification and fast prediction of behaviors

Alexander I. Hsu [1] & Eric A. Yttri [1,2 ✉]

Studying naturalistic animal behavior remains a difficult objective. Recent machine learning advances have enabled limb localization; however, extracting behaviors requires ascertaining the spatiotemporal patterns of these positions. To provide a link from poses to actions and their kinematics, we developed B-SOiD - an open-source, unsupervised algorithm that identifies behavior without user bias. By training a machine classifier on pose pattern statistics clustered using new methods, our approach achieves greatly improved processing speed and the ability to generalize across subjects or labs. Using a frameshift alignment paradigm, B-SOiD overcomes previous temporal resolution barriers. Using only a single, off-the-shelf camera, B-SOiD provides categories of sub-action for trained behaviors and kinematic measures of individual limb trajectories in any animal model. These behavioral and kinematic measures are difficult but critical to obtain, particularly in the study of rodent and other models of pain, OCD, and movement disorders.

[1] Department of Biological Sciences, Carnegie Mellon University, Pittsburgh, PA, USA. [2] Neuroscience Institute, Carnegie Mellon University, Pittsburgh, PA, USA. ✉email: eyttri@andrew.cmu.edu

The brain has evolved to support the generation of individual limb movements strung together to create natural behavior. The selection, performance, and modification of these actions is key to an animal's continued survival[1]. Establishing the neural underpinnings of this behavioral repertoire is one of the foundations of neuroscience[2]; however, research largely focuses on stereotyped, reductionist, and over-trained behaviors due to their ease of study. Beyond the potential confounds associated with artificial or over-trained tasks, this line of interrogation discards most of the behavioral repertoire and its intricate transition dynamics[3–5]. Comprehensive behavioral tracking requires accurate behavioral identification and quantification (e.g. kinematics and transitions at meaningful timescales).

Typically, behavioral scientists have relied upon top-down methods in which pre-established criteria are applied to behavioral data[6–9]. These methods, which include laborious human rating, have benefited from advances in supervised machine learning methods for classification[10], achieving accuracy on par with human labeling. Although these approaches can be useful, supervised machine learning classifiers are trained to replicate their user's annotations. These human annotations, however, are prone to observer biases and are known to suffer from high inter-rater variability[11–13] and typically possess low temporal resolution. Moreover, the experimental flexibility is typically quite limited. Because of their one-size fits all approach, the top-down rubric may have diminished sensitivity to the many perturbations that could not be encompassed in the training data set. However, these perturbations comprise the majority of use cases (see von Ziegler et al.[14] and Sturman et al.[13]).

To overcome these top-down limitations, Shaevitz, Berman, and colleagues began a new generation of unsupervised learning algorithms utilizing non-linear dimensionality reduction of the complex behavioral space to identify stereotyped behaviors (MotionMapper[15–17]). Specifically, movement is quantified by aligning the body in each frame, then extracting the spectral energy of the protruding limbs. This time-frequency information is then reduced down to a two-dimensional space (see Todd et al.[18] for review of various algorithmic implementations). The spectral energy component of this approach is particularly well-suited to extract the movement of orthogonal limbs, such as fly appendages sticking out from their bodies[19]. In soft-bodied invertebrates like worms and fly larvae, similar methods using decomposed body shape dynamics have been used with success[20–22]. However, these studies require model organisms that generate primarily orthogonal movements that are optimal for the frequency domain information they rely upon. As such, these methods have seen few applications in the study of vertebrate behavior. Additionally, to best extract this spectral information, these methods depend critically on a uniform background, commonplace in a fly dish, but more rare in vertebrate cages.

More recently, a proprietary package, MoSeq[23] advanced the field though the use of spinograms obtained from a specialized depth camera in conjunction with unsupervised hierarchical clustering methods to identify action groups. While MoSeq represents the first unsupervised segmentation in rodents, it highlights a greater issue concerning scales of behavioral extraction. First, both the action and its kinematics are critical, particularly to the study of several disease states[24–26]. Second, the low temporal resolution of most methods limits the applicability of any results with electrophysiological recordings. Third and perhaps most impactful, to maximize reproducibility and experimental efficiency, methods must be generalizable across sessions and across research groups. Current unsupervised methods are insufficient.

Recent advances in computer vision and machine learning have enabled automatic tracking of body part positions[12,27,28].

Although limb position or pose can be informative, its behavioral interpretability is quite low. For instance, the location of a paw may be used to determine stride length, but it does not capture what the animal is doing with that paw. Moreover, the various top-down frameworks that each user may create are incredibly subjective and may not generalize between animals of different sizes or cameras with differing frame rates[11].

Taking inspiration from the converging lines of technology, we created a platform that extracts the spatiotemporal patterns of these identified body poses (e.g. behaviors), of any subject. An important feature of our algorithm, B-SOiD, is that pose relationships are used to train a multi-class classifier that then can be used to bypass the intermediate transformation and clustering stages. In doing so, B-SOiD performs more quickly (100,000 frames/minute on a typical laptop) and with higher fidelity - as it is no longer limited to a single session's data set to define cluster boundaries. More importantly, once trained, the algorithm can generalize across animals, cameras, and setups, thus solving the issue of transference. With the utilized position information, B-SOiD provides a 2D readout of kinematics, and can provide temporal resolution in the single milliseconds, required for use with electrophysiological methods. We provide this platform as an open-source, step-by-step GUI interface to enable autonomous behavior identification and classification based upon the discovered pose relationships.

Here, we demonstrate B-SOiD's use in a variety of experimental models (mouse open field behavior, rat reach to grasp task, and human kinesiology data) We also benchmark the tool across different camera angles and against the current state of the art. Demonstration of distinct neural signatures corresponding to the identified behaviors and the analytical utility of the improved temporal resolution are also provided. Finally, we reveal robust kinematic changes following a cell-type specific lesion that are otherwise unobservable with current methods.

## Results

We provide here an open source tool to resolve distinct behaviors (Fig. 1). To achieve this end, we sought to make use of pose estimation software, which uses computer vision and machine learning to identify the location of body parts from video. These techniques have made huge strides in recent years, but making sense of those data remains difficult. We begin with a summary of the behavioral classification/segmentation tool (Supplementary Fig. 1), its computational underpinnings, and basic benchmarking. In addition to the extraction of behavior from poses, B-SOiD provides a signal processing method that provides temporal resolution matching the video frame rate. We then demonstrate the utility of this increased resolution - increased signal signal of the neural activity of behavior. In doing so we also provide neurophysiological verification of the mathematical-derived behavioral groups. We then quantify the algorithm's performance across different camera angles and compare it to the current state of the art. These measures also serve to validate the external and internal consistency of the method, respectively. The manuscript concludes with a real-world example of B-SOiD's potential, detecting several canonical grooming types and their kinematic composition, critical information that is not available via other methods.

B-SOiD is an openly available tool to identify and extract behavioral classes at millisecond timescales - all with a single, off-the-shelf camera (Supplementary Movie 1). Because B-SOiD identifies spatiotemporal patterns in labeled body part positions, it has no a priori limit on camera angle or organism (see Supplementary Fig. 2 and Supplementary Movie 2 for rat reaching task throughout training – including the same identified grasping

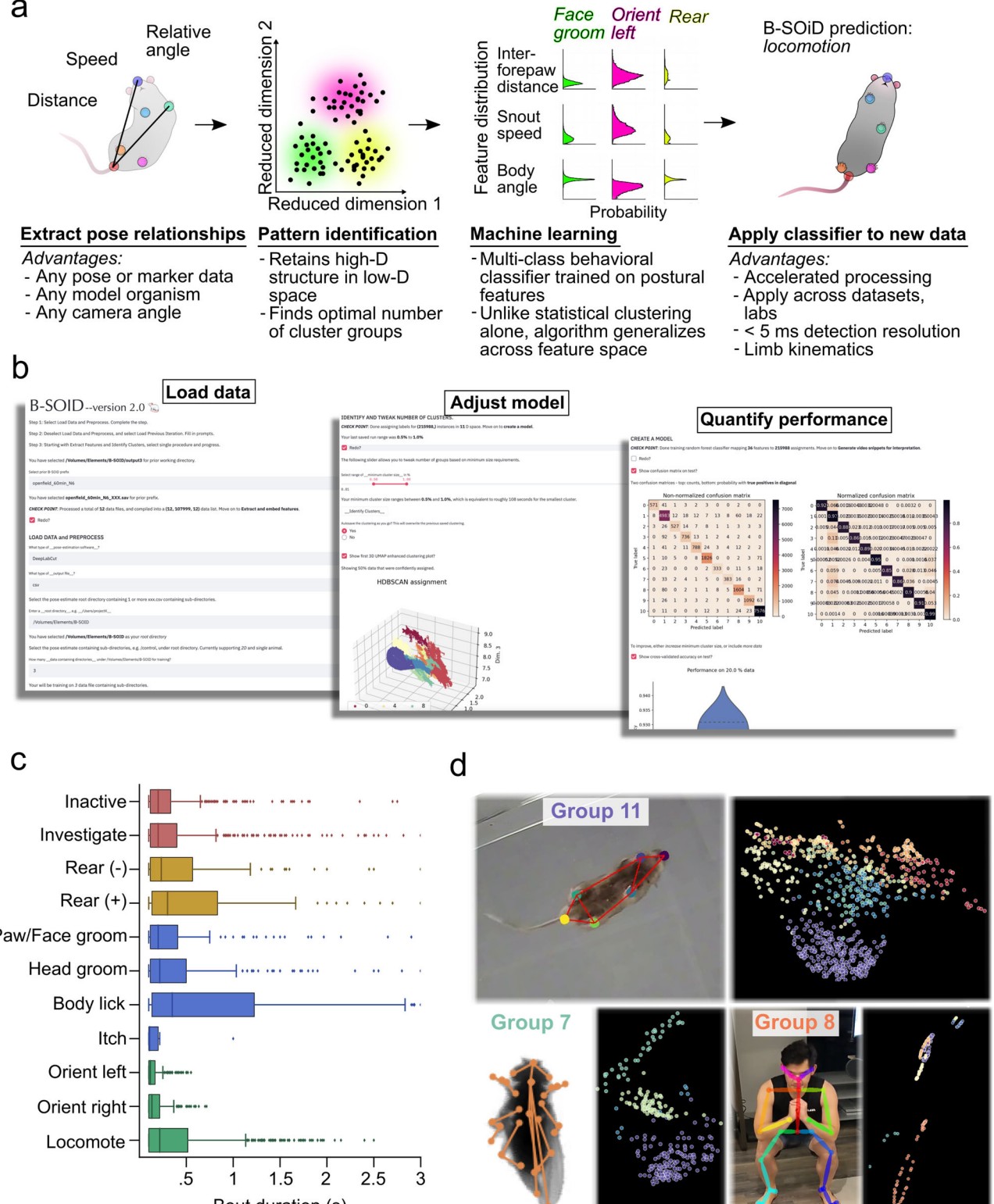

**Fig. 1 Summary of B-SOiD process, GUI, and performance in mice and other animals. a** After extracting the pose relationships that define behaviors, B-SOiD performs a non-linear transformation (UMAP) to retain high-dimensional postural time-series data in a low-dimensional space and subsequently identifies clusters (HDBSCAN). The clustered spatiotemporal features are fed as inputs to train a random forests machine classifier. This classifier can then be used to quickly predict behavioral categories in any related data set. Once trained, the model will segment any dataset into the same groupings. **b** Screen shots from B-SOiD app GUI, available freely for download. Examples of simple language progress from loading data, to improving model, and quantifying performance are shown. **c** Bout durations for each of the identified behaviors throughout an hour-long session, $n = 1$ animal. Data are presented as mean values ± SEM. **d** Snapshot of behavioral state space aligned to a freely moving mouse(DeepLabCut), fruit fly(SLEAP), and the first author - taken with a cell phone camera(OpenPose). Informed consent to publish the images was obtained. Color of group number refers to colored distribution within UMAP space. Source data are provided as a Source Data file for (**c**).

behavior with and without a sugar pellet present, Supplementary Fig. 3 and Supplementary Movie 3 for human exercising with positions extracted using OpenPose[29,30], and Supplementary Movie 4 for categorization of Drosophila behavior. Screenshots can be found in Fig. 1d for summary). For simplicity we focus here on a bottom-up view of six body part locations (snout, paws, tail-base, as identified by DeepLabCut) of a mouse in order to best resolve the animal's limb kinematics.

B-SOiD extracts the spatiotemporal relationships between all position inputs (speed, angular change, and distance between tracked points – Fig. 1a). After embedding these high-dimensional measurements into a low-dimensional space UMAP, a state-of-the-art dimensionality reduction algorithm[31], a hierarchical clustering method, HDBSCAN, is used to extract dense regions separated by sparse regions[32]. Although defining clusters in low-dimensional spaces is largely sufficient to achieve the desired behavioral identification[15,16,18,33], doing so is a computationally expensive process. Additionally, behavioral transference in the low-dimensional space is difficult to evaluate, owing partly due to the non-linearity in dimensionality reduction. To overcome both of these issues, we utilized a machine learning classifier that learns to predict behaviors based on the high dimensional measurements (Fig. 1a). This approach provides greatly improved computational speed (processing time for one hour of 60fps data containing six poses is under five minutes with a 128GB RAM CPU) and a consistent model that enables generalization across data sets within or across labs. Because the classifier is trained to partition pose relationships, not their low-dimensional representations, the defined clusters are further apart from one another, greatly improving consistency over statistical embedding methods (for unsupervised behavioral metrics comparing high vs. low-dimensional behavioral representation, see Todd et al.[18]. Finally, to improve functionality, we have increased accessibility – formatting the code into a downloadable app which provides an intuitive, step-by-step user interface (Fig. 1b).

### B-SOiD extracts behavioral clusters in high-dimensional space.

Behaviors can be parsed into a sequence of pose relationships that the brain has evolved to perform[34,35]. To reduce dimensions of those spatiotemporal pose relationships, B-SOiD implements Uniform Manifold Approximation and Projection (UMAP), next generation dimensionality reduction method[31] to simplify computations without simplifying the complexity of the behavior space. This non-linear dimensionality reduction approach provides an improved ability to delineate high-dimensional data in low-dimensional space over linear methods[31,36,37]. In particular, UMAP is preferred over t-SNE (t-Distributed Stochastic Neighbor Embedding) for its ability to preserve global pairwise distances in embedding. This feature is critical for users to manipulate behavioral delineation. More concretely, if the user considers segmented behaviors not critical for their research question, allowing preservation of global pairwise distances enables supervision in the number of behavioral groups.

Although non-linear dimensionality reduction algorithms may be advantageous when the output is two-dimensional, systematic exploration of unsupervised algorithms for animal behavior suggests that embedding in high-dimensional space improves results across various metrics[18]. To that end, we allowed UMAP embeddings to exist in a high dimensional space. Similar to Todd et al., we projected our data down to the number of dimensions required to achieve ≥0.7 variance explained using PCA. In this dataset assembled across six animals, the criteria number of dimensions was 11. To segregate behavioral assignments in the 11-dimensional UMAP space, we employed a hierarchical clustering method – Hierarchical Density Based Spatial

Clustering of Applications (HDBSCAN)[32]. Similar density-based clustering methods have been employed for unsupervised segmentation of behaviors in both vertebrates and invertebrates[15,16,38–42]. However, HDBSCAN is particularly well-suited to address the inevitable variability in pose estimations, even with with state-of-the-art software (see Methods section for specific HDBSCAN parameters), enabling B-SOiD to purify the training data to assign every frame.

### Algorithmic benchmarking.

When trained B-SOiD on video from six mice, in which it identified 11 classes in the 11-dimensional space (clearly distinguishable in pose relationship space, Figs. 1c and 2b, see https://github.com/runninghsus/bsoid_figs/blob/main/examples/README.md for spatiotemporal relationship distributions. Note, dimensionality count and group count are the same only be happenstance). Though not a given, the conserved kinematic motifs of the groups easily mapped onto established ethological names. For organizational purposes, we grouped these behaviors according to http://mousebehavior.org/ethogram-index/ (red = quiescence, gold = rear, blue = maintenance, green = move; to be used throughout this manuscript).

As a first pass to verify that B-SOiD did not errantly merge behaviors, we randomly isolated videos based on behavioral class assignments, and found behavioral assignments to be internally consistent (see https://github.com/runninghsus/bsoid_figs/blob/main/examples/README.md for details). This visual consistency approximates the human rating that a supervised algorithm would be based upon. In addition, meta-analyses on physical features showed distinct multi-feature distributions (full parameter distributions for each group available at the same link).

We observed that the clustered groups of spatiotemporal pose patterns did not change with animal color or size. The body length of the brown mouse in Fig. 2a is 50% greater than that of the black mouse (5.9 cm vs 3.9 cm), but both were clustered with the same B-SOiD model (and both contributed to the validation metrics shown here). We noted that in some instances B-SOiD ignored subtleties in body conformation, i.e. grooming at different places along the torso were all considered to be members of the same 'body lick' group (Fig. 2a). In other instances, B-SOiD separated related but fundamentally distinct kinematic patterns. In particular, without instruction to do so, B-SOiD identified the three canonical grooming types. These actions, first described decades ago as the syntactic chain of self-grooming in rodents, are paw/face groom, head groom, and body lick[34,43,44]. Itching with the hind leg was also identified, distinguished from any groom type using the forelimbs. This ability to both generalize and differentiate is vital to accurate behavioral segmentation and is largely the due to utilizing machine learning to recognize the spatiotemporal patterns. The algorithm seizes upon the conserved, repeated features and accepts the variability in others (see Supplementary Movie 1 for summary definitions and video examples). Note that cluster size limits can be adjusted in the aforementioned GUI, providing the user additional control over the grouping detail (see parsing of reach-to-grasp into sub-actions in Fig. S2).

To improve consistency, speed, and applicability in classifying behaviors, we equipped B-SOiD with a random forest classifier. The random forest classifier is well-suited for high-dimensional feature training and has been shown to predict low-dimensional representation of high-dimensional features well, particularly compared to potential alternatives like MLP or SVM[36] (see Methods section for classifier design). To test whether these pose relationships can be learned accurately, we tested the mapping on randomly selected 20% of the data. The predicted labels generated

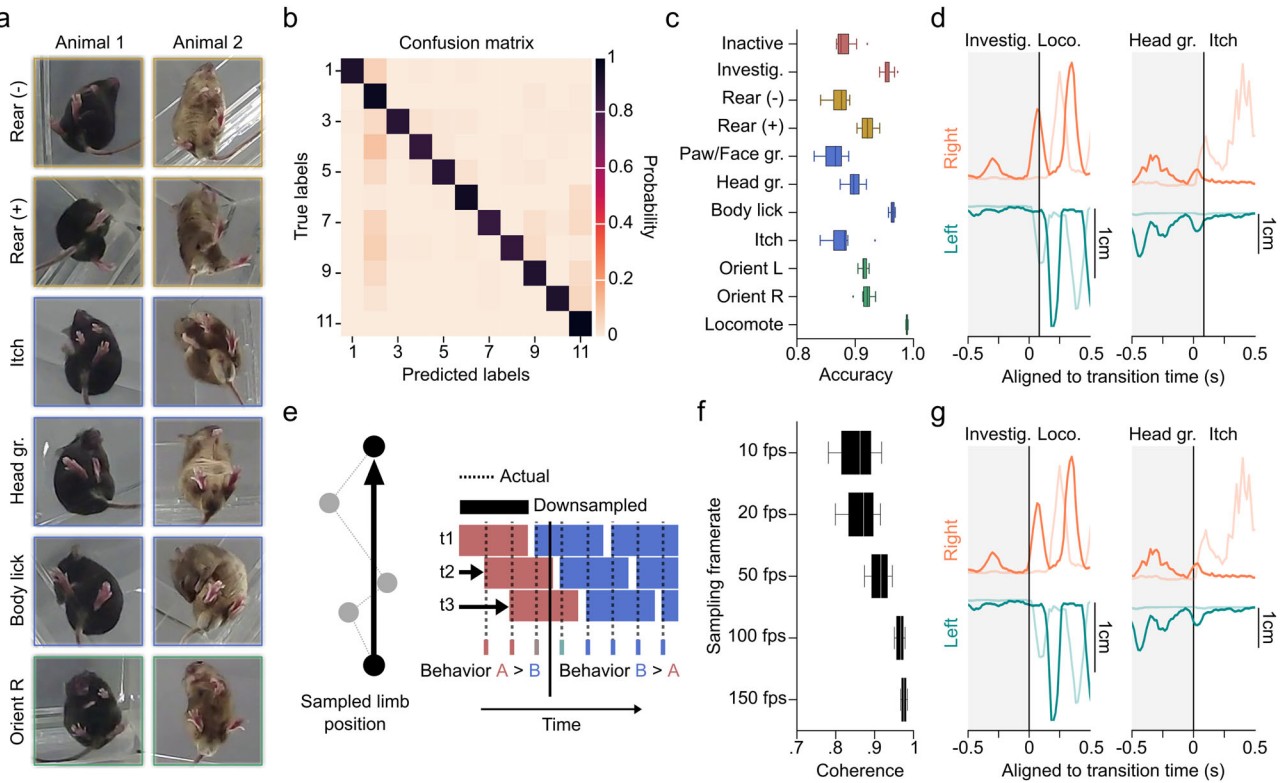

**Fig. 2 Performance quantification across multiple temporal resolutions with novel machine learning algorithms. a** Snapshots of a small (Animal 1) and large (Animal 2) mouse, 300 ms into the execution of example behaviors. **b** Confusion matrix on the 20% held-out data. True positive predictions appear on the diagonal. **c** 10-fold cross-validation yield high accuracy on shuffled data across behavioral groups (21,600 data points/test), n = 6 animals. Data are presented as mean values ± SEM. **d** Trajectory plots of right (orange) and left (teal) limbs of the fore paw (darker) and hind paw (lighter) demonstrating example transitions from investigate to locomote, and head groom to itch. Vertical lines denote transition time as identified by basic B-SOiD analysis (10fps). **e** Schematized example of the potential for prediction noise in pose estimation to override the movement signal at high sampling rates. To overcome this, we executed a frameshift computation to derive high resolution transition times from downsampled, high signal data. **f** Percent coherence between low frequency and progressively higher resolution frameshift data. A major break occurs under for data under 50 fps, n = 1 animal. Data are presented as mean values ± SEM. **g** Same trajectories as in (**d**), now incorporating the frameshift algorithm to improve resolution of transitions. Source data are provided as a Source Data file for (**b**, **c**, **f**).

by our random forest classifier matched cluster assignments by HDBSCAN ('true labels') over 90% of the time. Indeed, the confusion matrix and 10-fold validation indicate that high-dimensional features can be robustly assigned given low-dimensional group assignments (Fig. 2b, c).

**Frameshift paradigm enables behavioral segmentation at temporal resolution sufficient for electrophysiology.** Accurate resolution of the timing of behavior transitions is a necessary feature of segmentation beyond identification. We present two example transitions (Fig. 2d), at 10 frames per second (fps) a temporal resolutions on par with many current methods. Although the group identification is correct, the large inter-frame interval misses the transition time, leading to much of the behavior being inaccurately categorized. Resolving transitions with adequate precision for use with electrophysiological measures requires considerably faster sampling rates, which are unavailable given current technology. However, a particular challenge in defining behaviors at a high sampling rate is that pose location jitter dominates the signal from any movement (Fig. 2e, left side). It is precisely this loss of frame-to frame difference at high sampling rates that makes 10 fps sampling a popular temporal resolution.

To enable the resolution of behavioral transitions at the scale of single milliseconds, we introduced a 'frameshift' manipulation, borrowed from recent automatic speech recognition

innovations[45] (Fig. 2e). Briefly, B-SOiD initially downsamples all video, regardless of framerate, to 10 fps to achieve a high signal to noise ratio in the spatiotemporal dynamics of the markers. The process is then repeated, with each new set of predictions made on downsampled data, each time offset by an additional frame (see t1, t2, t3 in Fig. 2e right side). In essence, we decompose the high-resolution signal and run a sliding threshold for fitting the high-SNR decomposition. By combining behavior assignments extracted from the shifted, downsampled data, we gain improved transition time resolution while overcoming the hurdle of decreased signal-to-noise (Fig. 2f, g). Note, improving transition resolution does not fundamentally change the distribution of action durations observed at 10 fps. Thus frameshifting carries over the robust behavioral signal provided by lower sampling to the native resolution of the camera used.

As an example, B-SOiD automatically downsampled the 200 fps used here to 10 fps, then segments the downsampled data 20 times; each iteration offset by a single, 5 ms frame. We quantified the effect of different temporal resolutions by first subsampling a 200 fps video, thus providing an internally consistent comparison across resolutions (e.g. to extract 20 fps for analysis, we used every 10th frame of the 200 fps video). B-SOiD was then run independently on each resolution of the video, using frameshifting on each version. With the original 200 fps as a standard, we observed that the predictions across resolutions was highly coherent. Even at 10 fps (which was not

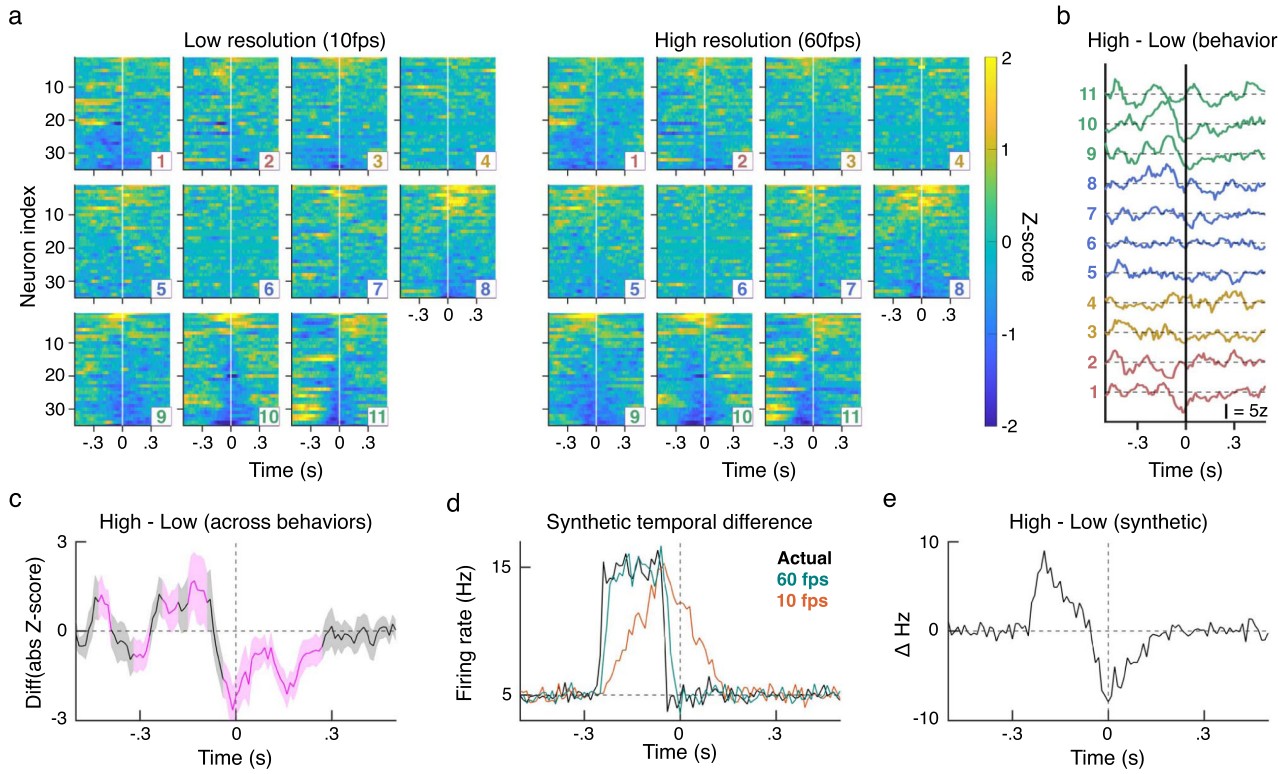

**Fig. 3 Frameshifted high-temporal resolution improves identification of neural signatures at behavioral initiation. a** Z-scored neural activity aligned to the 11 B-SOiD identified behaviors using either low temporal resolution, non-frameshifted or high-resolution, and frameshifted alignment. Neurons and neuron order are the same for each pair of behavior panels. Detailed plots can be found in Supplementary Fig. 5. **b** Total signal magnitude difference (high - low resolution) for each of the behaviors (1 on bottom). Scale bar = 5 z-score difference. Colors as in Fig. 1c. **c** Mean and SEM of signal magnitude difference across all behaviors (magenta = $p < 0.01$, two-tailed $t$-test). Positive values indicate greater signal magnitude for high vs low temporal resolution. **d** Using simulated data, we measured the average firing rate with zero (Actual), high resolution (60 fps), or low-resolution (10 fps) temporal jitter introduced. 60 fps produced a considerably more accurate account of the ground truth model. **e** Incorporating features from our recording data, the model produces similar high-low resolution difference dynamics to (**c**), here in a spiking artificial neuron. Source data are provided as a Source Data file for (**a**).

further downsampled/frameshifted), we observed a median coherence of ~84% across behavioral groups, Fig. 2f. As sampling rate increased, coherence improved – although the added benefit of increased sampling rate plateaus after 50 fps. These changes can be attributed to an increase in transition time accuracies, as seen in Fig. 2g and Supplementary Fig. 4. The frameshift paradigm allows B-SOiD to predict behaviors at a temporal resolution matching the sampling rate of the original video, enabling a notably deeper analysis of action kinematics (Fig. 2d, g). Given the excellent performance above 50 fps and the impetus to use less-specialized cameras, the remainder of this manuscript focuses on easily attainable 60 fps video.

**Increased transition fidelity improves neural signature resolution.** Improved temporal resolution is a critical advancement for analyzing neural correlates of spontaneous behaviors[46]. To assess the real-world benefit of increased temporal resolution, we simultaneously recorded 35 units from the left caudal forelimb area of motor cortex in a mouse as it navigated the open field arena (see Methods section). We then aligned the activity to the onset of classified actions using non-frameshift (10 fps) and frameshift (60 fps) predictions.

In this first demonstration of motor cortical activity aligned to the breadth of naturalistic behaviors observed, we noted distinct neural signatures for the range of identified naturalistic behavior groups (Fig. 3a). Across behavioral groups, these population representations were quite robust and observable with both high and low-resolution versions of B-SOiD (see Supplementary Fig. 5

for detailed account of neural activity by group). More importantly, these clear population responses indicate the mathematically established B-SOiD groupings reflect real distinctions at the level of neural representation. The strength of the aligned population responses largely coincided with actions involving the forelimbs, consistent with the recording location within motor cortex – although this simple descriptor cannot broadly summarize the diverse dynamics discovered (Supplementary Fig. 5). We also noticed a trend for greater modulation for orientations in the direction contralateral to the recording (group 10) compared to ipsilateral (group 9), although some neurons were preferentially modulated for ipsilateral orienting (Supplementary Fig. 6). While in-depth future analyses will be required to understand these responses, these data strongly support the quality of B-SOiD's clustering and its the potential for the study of the neurophysiology of unconstrained behaviors.

In addition to these neural correlates of spontaneous behaviors, we observed that frameshifted data yielded a greater magnitude of neural modulation. The improved neural resolution was particularly pronounced just before and during the time of each action's onset. To quantify these differences for each neuron, we subtracted the magnitude of the low-resolution activity from the magnitude of the higher resolution activity (e.g. positive values = stronger signal with high resolution frameshift method). Differences in signal quality across neurons and groups for individual neurons can be found in Supplementary Fig. 5. We then summed these within-session signal differences across all neurons, without any assumption whether a neuron was tuned to

that behavior (tuned neurons should contribute to the sum, while untuned should have zero net effect). Higher temporal resolution yielded improved signal preceeding behavioral onset across several actions groups (Fig. 3b) and on average across all segmented behaviors (Fig. 3c). Again, the largest differences were typically observed for actions involving forelimbs.

This improvement is considerable taking into account the only difference between the data sets is a 50 fps improvement in behavior onset resolution. The increase in signal strength yields an improved ability to detect more nuanced dynamics, and the duration of this improvement may be instructive as to the time course of motor planning in this population.

Several of the plots of the differences between high minus low-resolution demonstrate a biphasic dynamic in which the quality of the high resolution signal is initially greater, then worse, than the low-resolution signal. To better understand this dynamic, we modeled a simple neuron with a Poisson-distributed firing rate. This rate instantaneous increased from 5 Hz to 15 Hz. In our data and generally assumed for movement-related activity, we observed neural modulation occurring before movement onset and for relatively relatively short durations. Therefore the increase in synthetic activity was made 100 ms long and began 130 ms before onset. We then sampled the synthetic activity with 60 fps and 10 fps resolution onset jitter. Peri-event time histograms of the resulting signals demonstrate that 60 fps behavior resolution yielded dramatically improved results that were quite close to the zero jitter ground truth (Fig. 3d). The resulting difference in observed signal between methods was similar to that observed in the population activity (Fig. 3e). Specifically, we found the late improvement in low-resolution signal to be the result of a delay in resolving the cessation of the activity increase. This rudimentary summary demonstrates that B-SOiD's increased action alignment resolution prevents both signal degradation and temporal displacement of neural activity pattern.

**Comparison between top-down to bottom-up camera angles**. To optimally extract limb 2D kinematics (e.g. stride length, horizontal limb speed), we have focused on a bottom-up camera setup. This arrangement also provides an ideal situation for tethered animals, eliminating problems caused by the cable tether. However, many research groups prefer to use or have existing data from top-down cameras placed above a cage or arena. Additionally, a transparent floor may alter behavior or induce anxiety[47], and therefore may be suboptimal for some experiments. Using a session recorded simultaneously from above and below, we tested the performance of B-SOiD in different camera positions. For consistency, we used six points for the generation of both the bottom-up and top-down B-SOiD prediction models (for top-down: top of snout, shoulders approximation, hips approximation, and tail-base were used). In this head-to-head comparison, B-SOiD extracted eight behavior groups categories from top-down video. These groups largely mirrored with those identified with bottom-up video (sample ethogram Fig. 4a, c and category labels Fig. 4b, d; colors to group action types as in Fig. 2). No new actions were found and some related action groups were combined (e.g. elevated and lower rearing were combined into one group). Unsurprisingly, we did observe some divergence in groups that relied upon precise paw localization (e.g. grooming-type behaviors), which is difficult to achieve when viewed from above. In these cases of misalignment, assignments typically defaulted to the most similar behavior type given the same amount of head movement but no information about paw position (see full kinematic properties for each group for direct comparison https://github.com/runninghsus/bsoid_figs/blob/main/examples/README.md).

While the ethograms provide an overall sense of the quality of each method, we sought to determine the relationship between behavior segmentations using the different views. We determined the percent overlap for each view's action group, e.g. for frames identified as 'Orient Left', what is the relative distribution of bottom-up groups? After mapping each segmentation onto the target frames, we discovered that action groups are largely conserved between the camera angles, with top-down and bottom-up groups correctly mapping onto each other at a rate of several hundred percent more than would be expected given the baseline distributions, effectively removing the bias for behaviors that happen more often (Fig. 4b). Therefore, while identical segmentation between the two camera angles is impossible, we suggest that both approaches are valid and demonstrate high inter-method consistency.

**Comparison against alternative unsupervised pose-estimation method**. To benchmark B-SOiD against the state-of-the art in unsupervised behavioral segmentation, we compared the performance of B-SOID to MotionMapper on identical, bottom-up video data sets. The family of open source MotionMapper methods are the leading unsupervised method for behavior segmentation[18] and uses spectral information to discern behaviors. MotionMapper has a release that uses the same pose estimation input as B-SOiD (https://github.com/DeepLabCut/DLCutils/tree/master/DLC_2_MotionMapper), providing the means for a direct comparison. We made no assumption of ground truth for comparison; rather we focused our evaluation on the quality of the segmented behavior. We extracted and aligned frames of identical dimensions around the animal. For each bout we measured the motion energy of each pixel in the frames comprising that bout (the bright, constant background did not significantly contribute to motion energy), then computed the mean motion energy (ME) per behavioral group. Movement conserved across bouts will yield sharp and clear mean ME values. While the input data were identical, differences in the quality of groups were apparent. Compared to MotionMapper, summary images of B-SOiD groupings were more distinct from each other. Additionally, in several within-group panels, the average ME signal was clear enough that both limbs and group identities are visually apparent (Fig. 5a).

To quantify these differences, we computed the ME image mean sqaured error (MSE, see Methods section for details) for up to 20 randomly selected bouts per group. To reduce the effects of bout length, only bouts lasting 300–600 ms were used; thus occasionally groups with only very short bouts were under-represented. The unassigned noise group in MotionMapper was discarded for these analyses. The percent difference in MSE was then computed across in-group examples, providing an estimation of the in-group variability (Fig. 5b, in-group comparisons are contained within the diagonal, see Methods section for details). Normalizing to the mean in-group values, percent differences in out-group values were also obtained. Darker colors indicate a greater difference in ME. Although some in-groups may exhibit greater differences in ME than others (e.g. locomotion vs inactivity), well-clustered bouts should be more different from out- than in-group bouts. We then summarized the differences across all groups (Fig. 5c). We also provide results from the same B-SOiD bouts, shuffled into randomly assigned groups, thus providing a baseline for structureless in-group and out-group variability to be expected.

The extent of divergence of out-group MSE (dashed line) relative to in-group MSE (solid line), is indicative of the quality of groupings – specifically how different a group is from the remaining population. Both algorithms demonstrated significant

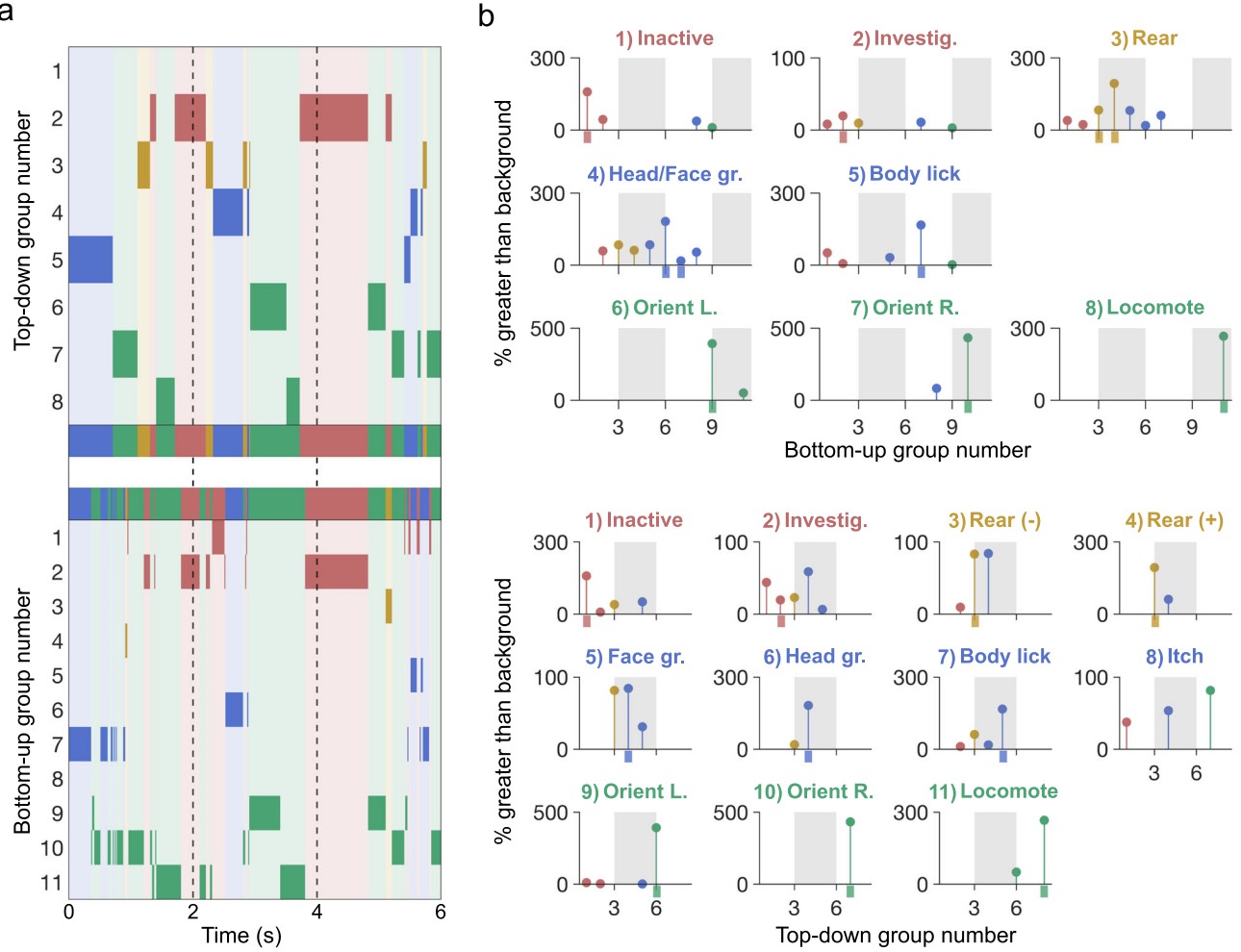

**Fig. 4 Comparison of top-down to bottom-up camera angles. a** Example ethograms of concurrent behavioral segmentation from the top-down view. The group definitions for the behavioral groups are as titled in (**b**), Fig 1c. **b** To quantify the relationship between approaches, top-down video reference groups were mapped onto bottom-up target groups. Y-values indicate percentage of overlap greater than baseline distribution. Values < 0 not shown. **c**, **d** Same as (**a**, **b**), but using bottom-up as reference and top-down as target. Tick mark along *x* axis indicates correct target group. Source data are provided as a Source Data file for (**a**–**d**).

rightward shifts of their out-groups, but the effect was much more pronounced in the B-SOiD data (MotionMapper: $p < 3e\text{-}12$; B-SOiD: $p < 7e\text{-}111$; Shuffled: $p = 0.60$). MotionMapper has constituted a pivotal advance, opening the door to unbiased analysis of the richness of unconstrained behavior. We recognize that the strength of the method lies in the ability to process spectral data from organisms with body components moving orthogonally to its center of mass, such as fruit flies. Thus, this pose-adapted method may benefit from greatly increasing the number of body positions identified, effectively providing similar spectral information. Finally, in the comparison of methods, we note that aspects MotionMapper are memory-limited, leading to roughly a 100X difference in processing time compared to B-SOiD with only six points. More points will exaggerate these differences. Some of this two-order of magnitude differences can be attributed to our integration of the novel UMAP technology rather than t-SNE (https://umap-learn.readthedocs.io/en/latest/benchmarking.html)

**Robust and often unmeasurable kinematic changes resolved with B-SOiD.** To assess B-SOiD's real-world utility, we quantified grooming-type behaviors in mice with and without cell-type specific lesions of the indirect pathway of the basal ganglia (A2A-

cre, with or without cre-dependent caspase virus injected into striatum, $N = 4$ mice each; Supplementary Fig. 7). The basal ganglia is thought to be involved in action selection and sequencing[34,46], the dysfunction of which may give rise to diseases like OCD and Huntington's, in which unwanted actions occur, or occur too quickly[48]. Additionally, activation of the indirect pathway has been suggested to contribute to hypokinesis, or smaller and slower actions[49]. Importantly, methods for measuring these kinematic changes (e.g. limb speed and distance) across behaviors are largely absent, aside from locomotion. We first compared the individual strokes comprising bouts of head and face grooming.

Consistent with a hypokinetic role, we found that across all animals lacking indirect pathway neurons there was a significant rightward shift in the speed and distance of face grooming, particularly pronounced for the smaller movements in the distribution (Fig. 6). However, these effects was not observed in the similar, but generally larger head grooming behavior. Current quantitative methods for grooming only provide bout duration and typically the canonical grooming types are combined because of technological limitations. Realizing these robust but hitherto indiscernible effects is made possible because of B-SOiD's ability to dissociate groom types, measure kinematics, and accurately identify the start and stop of bouts. We were also able to uncover

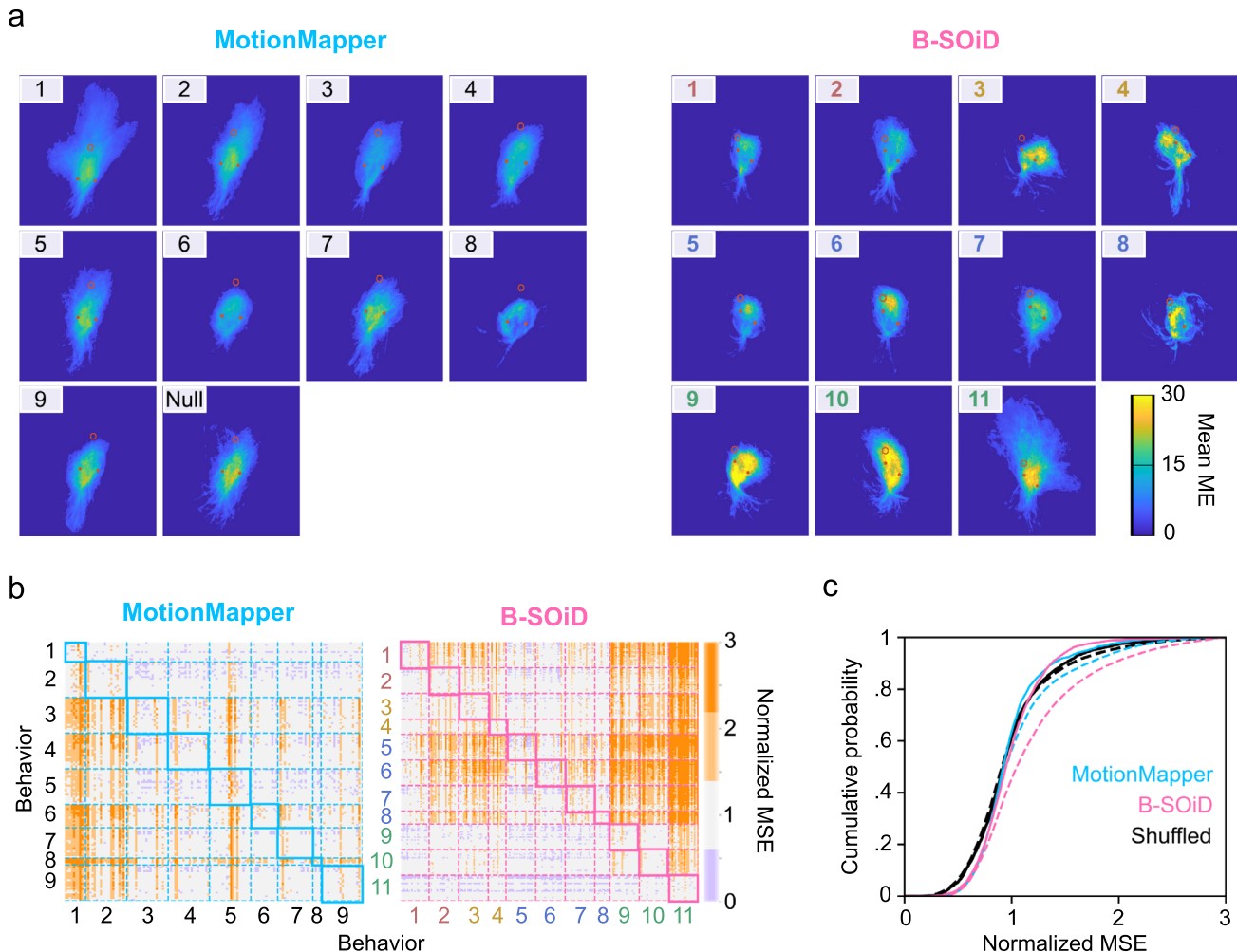

**Fig. 5 Quantification of unsupervised segmentation algorithms. a** Using the same pose-estimation data, we selected up to 20 bouts from each behavioral group identified by either DLC_2_MotionMapper or B-SOiD to construct motion energy (ME) images - capturing the average amount of movement across bouts. Brighter colors indicate greater consistency in movement over the 300-600ms bouts. **b** To quantify the quality of these groupings, we determined the difference (MSE) in motion energy across every bout, normalizing every values along a row to that row's in-group mean MSE. The comparison between related, in-group bouts is shown in the highlighted diagonal. Darker orange indicates greater differences between those pairs of bouts, e.g. 2 = twice the normalized MSE. B-SOiD behavior numbers as used in Fig 1c. **c** Cumulative histograms of values in (**b**) for in-group (solid line) and out-group (dashed line) bouts. The same B-SOiD bouts were shuffled into 11 random groups (black) to demonstrate a distribution without structure. Right-shifting of the distribution is indicative of increased differences between sample bouts. Source data are provided as a Source Data file for (**b**, **c**).

other kinematic effects (Supplementary Fig. 8), including pronounced increases in itching speed. Locomotor stride length, but not stride speed was also significantly increased, providing a kinematic mechanism for previous seminal motor control work that observed gross locomotor hyperactivity following indirect pathway lesion[50]. The behavior-specific kinematic sensitivity demonstrated here may provide the means to uncover deeper understanding in the fields of motor control, OCD, and pain[48,51,52].

## Discussion

Naturalistic, unconstrained behavior provides a rich account of an animal's motor decisions and repertoire. Until recently, capturing these movements with precision and accuracy was prohibitive, as evidenced in part by the relative lack of computational ethology studies[53]. Still, position does not equal behavior. Rather, it is the stereotyped spatiotemporal patterns of these positions that yield behavior. Our unsupervised algorithm, B-SOiD, captures the inherent statistics of limb and action dynamics with off-

the-shelf technology and a simple user interface. This tool serves as the vital bridge between recent breakthroughs in establishing the position of body parts[12,28] and the conserved patterns of positions we call behaviors. It also demonstrates the utility and potential of pairing unsupervised spatiotemporal pattern extraction with supervised machine learning classifier in behavioral assessment. It is the patterns in pose relationships that are discovered, extracted, and used to inform the ML classifier, thus mathematically tailoring the tool to the subject's inherent behaviors and eliminating top-down user bias.

In addition to providing a tool that can be used on any position data, we provide a glimpse into its potential. In part enabled by the improved temporal resolution, we were able to align action onsets with cortical activity, uncovering neural dynamcs that reflected the changing behaviors. The greater resolution also provides increased sensitivity to detect brief behaviors and the individual components of those movements. In particular, we resolved kinematic changes of individual limb strokes (grooming, itching, locomotion) in a lesion model. The ability to decompose

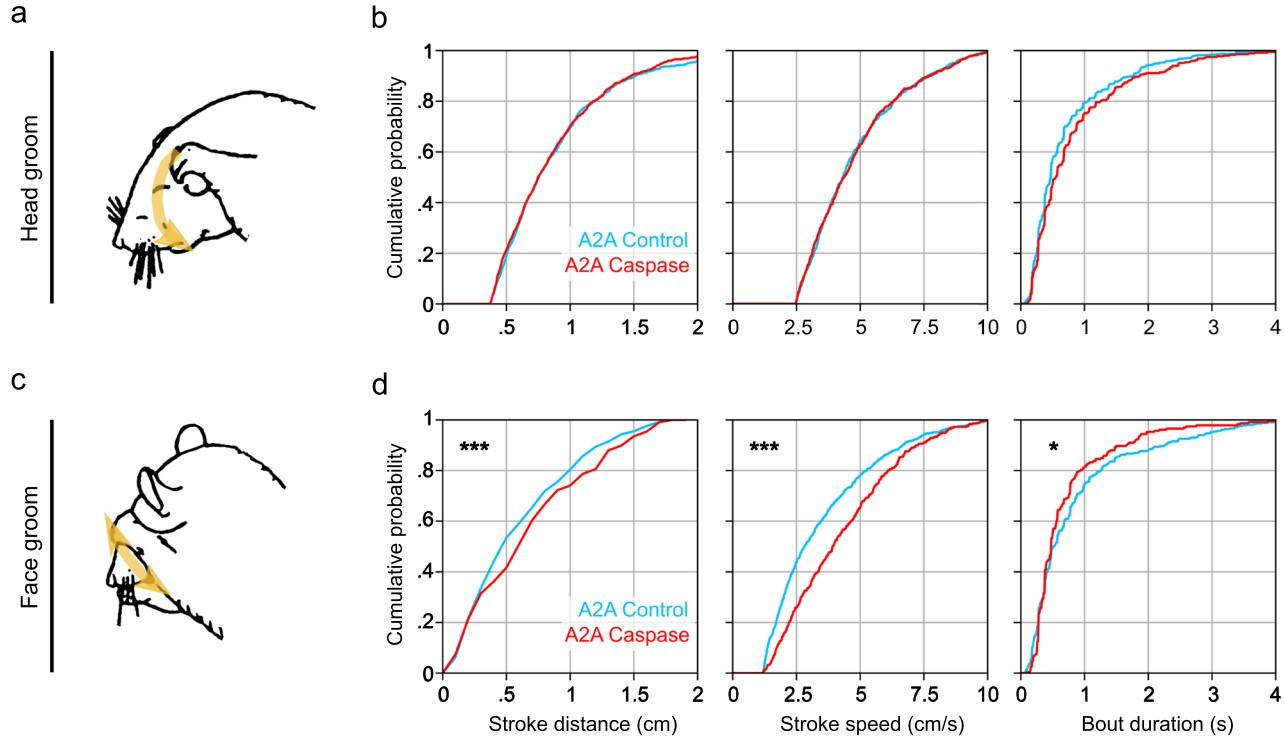

**Fig. 6 Detection of robust, hard to detect kinematic changes in grooming behavior following cell-type specific lesion. a** Schematic of canonical head grooming and (bottom) face. **b** Cumulative histograms of distances and speed of the individual strokes comprising a bout, and of bout duration (control - blue; lesioned - red, N = 4 animals, one session each). *p < 0.05, ***p < 0.0001, two-tailed Kolmogorov-Smirnov test). (**c**, **d**) Same as (**a**, **b**), but for canonical face grooming. Source data are provided as a Source Data file for (**a**).

behaviors into their constituent movements is a key feature. Because B-SOiD uses limb position, it extracts not only the action performed, but also kinematics (stride speed, paw trajectory, etc). While recent work has benefited from access to such performance parameters[26,54], it stands to be an even more potent advantage in the study of disease models. Obsessive compulsive disorder research in particular has long sought improved identification and quantification of grooming behavior[43,55,56]. Pain and itch research has also sought to achieve similar ends[52]. These results point to the need for a deeper comprehension of the composite kinematics forming those actions, as many current methods are limited to only the duration of such actions.

A unique advantage of the classifier built on these pose-relationships is computational ease. First, the open source package provides a platform that is accessible to biologists without extensive coding knowledge nor computational resources (e.g. expensive GPUs). Next, the classifier provides greater flexibility, allowing a single trained model to be generalized across subjects, labs, or frames with minor positioning errors. This ability is fundamental deliverable of the machine learning integration. The random forest classifier is trained on pose relationships, extracting the conserved, essential features while also recognizing those features with high variability. The classifier then predict the likeliest label given all of the features, eliminating the potential concerns that are inherent in non-linear transformations. Finally, the speed afforded by using a classifier, rather than clustering, to segment behavior leaves open the door to perform closed-loop manipulations through real-time segmentation[57].

To discover the clusters on which the classifier is built, B-SOiD uses UMAP, which is both more computationally effective and faster than similar methods like t-SNE[31]. While both methods presetve tje local structure, UMAP preserves the "global structure", or long-distance embedding placement of the dimensionally reduced space. Said another way, UMAP can enables the

determination of whether point 1 closer to point 2 or 3. These advances may be useful for future development and interpretability. The improved stability and speed of pose to re-embedding enables segmentation at speeds greater than most camera framerates. Additionally, B-SOiD need not be restricted to segmenting behaviors based only on pose-relationships. High-dimensional, unsupervised segmentation may also be able to integrate multi-model signals such as acoustics, environmental stimuli, or multi-animal social interactions.

## Methods

Here we explain the in vivo behavioral and electrophysiological measurements, and then proceed to the computational techniques underlying the algorithm and further analyses. A processing diagram can be found in Supplementary Fig. 1. For bout duration in Fig. 1c, model cross-validated accuracy in Fig. 2c, and frameshift coherence in Fig. 2f, box-plot elements carry the standard definitions(e.g. center line, median; box limits, upper and lower quartiles; whiskers, 1.5x interquartile range; points, outliers)

**Behavioral subjects and experimental set-up.** For normal, non-lesion experiments (Figs. 2, 4, and 5), subjects were six, adult C57BL/6 mice (3 females, Jackson Laboratory). The single brown mouse shown in Fig. 2a, is one of those six mice, and is an example of the diversity of the C57BL/6 line, specifically the substrain 6N (NIH) lineage, which can produce brown fur. Individual animals were placed in a clear, 15 × 12 inch rectangular arena for one hour while a 1280 × 720p video-camera captured video at 60 Hz (cluster and ML) or 200 Hz (frameshift). Following each data session, any feces were cleaned out and the arena was thoroughly sprayed and wiped down with Virkon S solution. The arena was then allowed to dry and air out for several minutes. This video was acquired from below, 19 inches under the center of the field. Offline analysis was performed in either Python or MATLAB (MathWorks). Unless otherwise mentioned, all statistical measures of behavior were non-parametric, two-tailed Kolmogorov–Smirnov tests and all error bars are the standard error of the mean. All animals were handled in accordance with guidelines approved by the Carnegie Mellon Institutional Animal Care and Use Committee (IACUC).

**Electrophysiology.** To demonstrate differences in temporal resolution (Fig. 3), we recorded during one open field session from layer 5 of forelimb area of motor

cortex (0.50 mm anterior, 1.75 mm lateral of Bregma, $z = 950 \mu m$) in one Drd1a-cre x ai32 on C57BL/6 background female mouse using a 64-channel silicon electrode chronically implanted in aseptic conditions (Cambridge Neurotechnology). Signals were sampled at 30 kHz with Open Ephys hardware. Spikes were high-pass filtered and sorted offline using Kilosort2 (https://github.com/MouseLand/Kilosort2). Activity across bouts aligned to onset of each behavior. To enable comparison across neurons, each neuron's average activity was binned (10 ms) and z-scored over the interval −1s to 2s relative to onset. Neural activity in Fig. 3 was z-scored across all instances of that action and are rank ordered according to the average low-resolution activity during the 200 ms prior to alignment time. This rank order was used for both panels of a given behavior. The neurons in Supplementary Fig. 5 are not rank sorted, but are consistent across panels so as to facilitate comparison across panels. Similarly, z-scoring across all panels incorporated all bouts identified with at the given resolution. A sliding boxcar with a semi-width of three bins was applied to activity visualizations. Randomly sampled data was achieved by taking the same number of instances of each action within the session and distributing this number of alignment times throughout the session. PSTHs were then generated with those alignments in the same manner (Supplementary Fig. 4).

**Extraction of kinematics.** Individual strides and grooms were identified, and the speed of the stroke quantified, by using the MATLAB function findpeaks() on the speed of the right forepaw, with the troughs serving as the start/stop of each movement. Average groom distance was computed as the euclidean distance of right forepaw displacement per stroke averaged amongst all head groom bouts.

**Indirect pathway cell-type specific lesion experiment.** Eight adult Adora2a-cre (often called A2A, Jackson Laboratory stock #036158, four females, C57BL/6 background) mice were used to study the effects of cell-type specific lesion of striatal indirect pathway neurons. Half of these animals (two females) were injected with AAV2-flex-taCasp3-TEVP[58] $4 \times 10^{12}$ vg/mL from UNC Vector Core bilaterally into the dorsomedial striatum: AP +0.9, ML ± 1.5, DV −2.65. Animals were allowed to recover at least 14 days prior to open-field experiments. The virus is designed to kill only cells expressing cre recombinase. To help visualize virus spread, a non-Cre dependent GFP virus, AAV2-CAG-GFP $4 \times 10^{12}$ was co-injected. 1 μL in each hemisphere was injected with a virus ratio of 2:1, Casp:GFP and a rate of 200 nL/min. The GFP virus was added because the cre-positive cells will be killed as a result of the caspace virus injection, leaving the many cre-negative cells behind (Supplementary Fig. 7). Thus the location of gross cell loss is difficult to quantify otherwise (GFP – green, foxpl1 counterstain – red, overlap – green). Expression of GFP is restricted to the striatum. Quantifying decreased cell density by eye also produced similar lesion maps (not shown).

**DeepLabCut training and model availability.** For pose-estimation, we used the aforementioned six-body part model trained on a total of 7881 frames (at least 50 frames for all sessions, 69 total sessions, $N = 21$ animals). The training regimen was set to the DeepLabCut default[12] and trained for 1.03 million iterations, achieving a loss of ≈0.002. The weights of the neural network are open sourced and freely available https://github.com/YttriLab/B-SOID/tree/master/yttri_bottomup_dlc-model/dlc-models.

**Data processing feature extraction.** With increasing sampling frequency, the intra-frame differences that are critical to determining the spatiotemporal features (e.g. speed) diminish. For instance, 60 fps sampling provides an inter-frame interval of only 16.7 ms – relegating the changes in position to a similar magnitude to the jitter in the position signal itself. To improve the signal-to-noise ratio, B-SOiD downsamples all input to non-overlapping 10 fps (100 ms) windows, and then either sums (displacement, angular change) or averages (distance) over all 10 fps samples. Thus, for the six points used in our mouse data, the per-frame spatiotemporal features consisted of 15 displacement (**D**) and angular change (**Θ**) measures, and six distances (**L**). This process is described in Algorithm 1 and the process pipeline diagram (Supplementary Fig. 1). In addition, these features are then smoothed over, or averaged across, a sliding window of size equivalent to ~60 ms (30 ms prior to and after the frame of interest). This is important for distinguishing the pose estimate jitter from finer movements that the animal makes, such as the different groom types.

## Algorithm 1

Feature extraction for $N$ pose estimates
Initialize, for $m = 1$ to $(\frac{N}{2})$:
$\mathbf{L}_m \leftarrow 0$
$\mathbf{\Theta}_m \leftarrow 0$
**for** $m = 1, M$ **do**
 $m \leftarrow$ any pair of pose $n$ and $\neq n$
 Store $\|(n_{m1}, n_{m2})\|^2$ in $\mathbf{L}_m$
 **for** $t = 1, T - 1$ **do**
 Store $\arccos [(\mathbf{L}_{m,t+1} \times \mathbf{L}_{m,t})/(\|\mathbf{L}_{m,t+1}\| \cdot \|\mathbf{L}_{m,t}\|)]$ in $\mathbf{\Theta}_m$
 **end**

 Discard the first index of $\mathbf{L}_m$
**end**
Initialize, for $n = 1$ to $N$:
$\mathbf{D}_n \leftarrow 0$
**for** $n = 1, N$ **do**
 $n \leftarrow$ 2D pose estimate
 **for** $t = 1, T - 1$ **do**
 Store $\|(n_{t+1}, n_t)\|^2$ in $\mathbf{D}_n$
 **end**
**end**
**return** $L, \Theta, D$

**Dimensionality reduction with UMAP.** B-SOiD then projects the computed pose relationships (**D**, **Θ**, and **L**) into a low-dimensional space, which facilitates behavioral identification without simplifying the data complexity. In simpler terms, similar mouse multi-joint trajectory will retain its similarity visualized in the low-dimensional space. B-SOiD achieves this through UMAP, a state-of-the-art algorithm that utilizes Riemannian geometry to represent real-world data with the underlying assumptions of the algebraic topology[31]. UMAP is chosen over the popular t-SNE for its advantage in computational complexity, outlier distinction, and most importantly, preservation of longer-range pairwise distance relationships[31,36,59–61]. Dimensionality reduction was implemented in the 'Start clustering' step in B-SOiD UI. Embedded in this step is a python implementation of umap-learn v.0.4.x (https://github.com/lmcinnes/umap). Since our goal is to use UMAP space for clustering, we enforced the following UMAP parameters: (n_neighbours = 60, min_dist = 0.0, euclidean distance metric). In terms of n_components, we call python implementation of decomposition.PCA() from scikit-learn v.0.23.x (https://github.com/scikit-learn/scikit-learn) and set n_components to explain ≥0.7 of total pose-estimation variance.

**Identify group assignments with HDBSCAN.** UMAP embeddings were then clustered through HDBSCAN algorithm[32]. It is particularly useful for UMAP outlier detections as it recognizes subthreshold densities. HDBSCAN assignments was implemented in the 'Start identifying' in B-SOiD UI. Embedded in this step is a python implementation of hdbscan v.0.8.x (https://github.com/scikit-learn-contrib/hdbscan). To enable maximum flexibility in determining the number of behavioral groups the method creates, we enabled user input for HDBSCAN parameter min_cluster_size.

**Random forest classifier for accurate and fast prediction.** Random forest classifier design was chosen for high-dimensional pose relationships mapping to discrete multi-class behaviors. In addition, it has been suggested that Random forest has the ability to accurately learn the low-dimensional embedding from the high-dimensional features[36]. Random forest was implemented in the 'Start training a behavioral random forest classifier' step in B-SOiD UI. Embedded in this step is a python implementation of ensemble.RandomForestClassifier() from cikit-learn v.0.23.x (https://github.com/scikit-learn/scikit-learn). We set the parameters to default, as it was sufficient for learning the mapping.

**Frameshift prediction paradigm.** Many end users may wish to apply the algorithm to higher frame-rate video. Because B-SOiD applies a temporal constraint of ~10 fps to maintain an optimal signal-to-noise ratio (which can be adjusted by tricking the UI in input frame-rate), we designed B-SOiD to predict along a sliding window. This is mathematically implemented using offsets, pseudocode in Algorithm 2.

## Algorithm 2

Frameshift implementation for $F$ times higher sampling rate than 10 fps
Initialize behavioral array:
$\mathbf{G} \leftarrow 0$
Initialize downsampled behavioral array, for $f = 1$ to $F$:
$\mathbf{g}_f \leftarrow 0$
**for** $f = 1, F$ **do**
 Start at $f$, sample pose-relationships at 10 fps, $S$ frames
 **for** $s = 1, S$ **do**
 Store the prediction $(\mathbf{g}|s)$ in $\mathbf{g}_f$
 **end**
 Insert $\mathbf{g}_f$ at every $F^{th}$ position in $\mathbf{G}$ starting at $f$
**end**
**return** $G$

In Fig. 2f, to accurately quantify the consistency between predicted frameshifted $\mathbf{G}$ and the non-frameshifted $\mathbf{g}_f$ (annotation described in Algorithm 2), we upsampled $\mathbf{g}_f$ with the same values prior. To demonstrate the input flexibility of B-SOiD with a high speed camera, our frameshift example in Fig. 2f was 200 fps. Given no evidence for improvement beyond 50 fps, all other analyses were 60 fps at resolution.

We recognize that spurious labeling could arise due to jitter. Although segmentation is applied independently to each frame, we assume some level of continuity of actions. As such, we discarded any frameshifted bout that was not at

least three samples (<50 ms) in duration where applicable (Figs. 3 and 4; Fig. 5 used a 300 ms lower bound and when set lower, the inclusion of shorter durations only improved B-SOiD relative to MotionMapper; the data in Fig. 6 was devoid of <50 ms bouts). In Fig. 3, non-frameshift was comprised of 10,433 bouts with a mean of 948 bouts/group; frameshift was comprised of 17,129 bouts with a mean of 1557 bouts/group. Given 11 groups, the probability of the random jitter event occurring and being repeated three times is less than one percent. We note that the addition of this conservative approach did not qualitatively change any findings, and that each datapoint itself is the summary of several segmentations across the 100 ms frameshift window. Although this duration cutoff is in line with other mouse behavioral work, our motivation was more mathematical and less a statement on any psychophysical assumptions. As demonstrated in the 300 fps rat reach-to-grasp data, B-SOiD as a tool can be titered to preferentially segment whole actions or their sub-actions. Additionally, some actions (e.g. saccades during reading) have durations as low as 20 ms. A human can deliver as many as 20 punches in a single second. As such, B-SOiD does not enforce any limits on duration, allowing the user to determine the pertinent timescales and appropriate interpretations.

**Integration of low-confidence pose estimates**. The occlusion of a point from view can be informative (e.g during a rear, the snout is often occluded by the body when viewed from below), but a missing point can be the result of poor pose estimation on a given frame. The certainty of each frame's pose estimation is provided by most pose estimation software in the form of prediction confidence/likelihood values. In all data sets, we observed a bimodal distribution comprised of either very high or low confidence values. To bisect the two distributions with each session, B-SOiD designates all points with a confidence score below the elbow point (difference (high - low) between adjacent likelihood becomes positive) of the probability. We remove these low-confidence points and substitute that position with last high-confidence position. Thus, the displacement between frames for a low-confidence point is zero. It should be noted that even when stationary, pose estimation programs do not output identical positions in consecutive frames.

Low confidence estimations can occur from a missing body part (e.g. from being behind another body part) or poor prediction (e.g. blurry video or inconsistent lighting). In the latter case, the aberrations which led to the low-confidence points are typically short lived, often only a single frame, and are fully mitigated by averaging over the 100 ms frameshifting interval. Prolonged low-confidence points contribute a spatiotempotal signature, and if repeated as a behavior, may be part of an identified spatiotemporal pattern (either in training or prediction). The occlusion of the snout during rear(+) is one example. Importantly, occasional prolonged aberrations do not adversely affect the algorithm. During training, spurious omissions will be too variable to constitute a conserved pattern – and the added variability may make the random forest even more robust. During prediction, the trained model utilizes 36 spatiotemporal features, minimizing the effect of a pose estimation error. This ability to incorporate patterned omissions and overcome spurious errors is another benefit of adding the trained classifier following the clustering.

**Motion energy image mean-squared-error**. The term 'motion energy' was introduced by Stringer et al.[62] and refers to the absolute value of the difference of consecutive frames. Since the animal is freely moving in the environment, starting pose alignment is necessary. Following image registration using estimated outline of animal at the start of each identified behavior, we compute the motion energy (ME, absolute value of the difference of all consecutive frames) using MATLAB command matlabimshowpair, capped at 600 ms for conciseness. We then performed weighted averaging for each bout to reconstruct a single ME image. In other words, each pixel in such reconstructed ME image represents the average absolute difference between consecutive frames at a given pixel location. Since there are multiple instances of each action, we want to see if such animal-centric average absolute difference is conserved between instances. To quantify consistency, we performed all pair-wise image mean-squared-error using MATLAB command matlabimmse. Essentially, the pixel difference between instances (ME images) will be coalesced into a single value (MSE). MSE is inversely proportional to consistency of animal movement for each identified action.

**Reporting summary**. Further information on research design is available in the Nature Research Reporting Summary linked to this article.

## Data availability
The data generated in this study have been deposited in a GitHub repository found at https://github.com/YttriLab/openfield_data, https://doi.org/10.5281/zenodo.4900573[63]. Source data are provided with this paper.

## Code availability
Our DeepLabCut network, analysis code, as well as the data used to create these figures, are all open sourced and freely available from GitHub https://github.com/YttriLab/B-SOID, https://doi.org/10.5281/zenodo.485010[64], including a similar version of the code

for MATLAB, though this version is simpler and uses t-SNE for dimensionality reduction.

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

## Acknowledgements

We would like to acknowledge to acknowledge Mary Cundiff and Aryn Gittis for providing A2A lesion animals. Reaching data acquired by Alexandra Bova in the laboratory of Daniel Leventhal (University of Michigan and VA Ann Arbor Health System). Talmo Pereira, Josh Shaevitz, and Mala Murthy provided fly data, tracked using SLEAP (sleap.ai). We also thank Alexander Hodge, Mark Nicholas, Nahom Mossazghi, and Andrew Buzza for data processing assistance, Andreas Pfenning and Sarah Ross (University of Pittsburgh) provided insightful comments, and Parley Belsey for art assistance. This work was supported by the Brain Research Foundation Fay/Frank Seed Grant and the Pennsylvania Department of Health.

## Author contributions

A.I.H. acquired and analyzed the data, conceived the algorithmic advancements, created the software, and drafted the manuscript. E.A.Y. conceived the study, analyzed and interpreted the data, and drafted the manuscript.

## Competing interests

The authors declare no competing interests.
