## [Peer Review File · Nature Communications]

B-SOiD, an open-source unsupervised algorithm for identification and fast prediction of behaviorsREVIEWER COMMENTS

Reviewer #1 (Remarks to the Author):

This manuscript by Hsu and Yttri describes a new approach for quantifying animal kinematics, clustering pose information to identify specific behaviors and how they change across time. While there have been huge strides in pose estimation from video in recent years, the next level remains a challenge: what should we do with the pose data? How can it inform us about how the animals are behaving across time? The authors describe a clustering approach called BSOiD, which identifies specific behaviors in videos that have been run through pose estimation using DeepLabCut or OpenPose. They show that they can identify “clusters” in these videos that correspond to specific behaviors. While there is a lot to like in their approach, the manuscript suffers from some deficiencies that should be fixed before publication.

1) The authors claim to demonstrate B-SOiD’s use “in a variety of experimental models (mouse open field behavior, rat reach to grasp task, and human kinesiology data)”. However, for two of these applications (rat reach to grasp task and human kinesiology data) they show almost no data, only a supplemental figure or video without description of what was being shown or concluded from these. It was a bit easier to see in the human example that they were identifying different poses but I was confused about the take home point of the rat reaching figure, or how to interpret the figure as this was not well described in the text. Was the inclusion of these just to demonstrate that they can run B-SOiD on poses extracted from videos other than mouse open-field? Did they learn anything about rat reaching or human exercise through these analyses? As it stands, I don’t think it’s fair to claim that the use of B-SOiD has been demonstrated in rats and humans, as the examples they show are anecdotal.

2) There appeared to be a tension in the paper between wanting to provide a simple tool that could be used cheaply and easily by many people with off the shelf cameras, vs. a high resolution tool that could be used for synchronizing pose information with electrophysiology or other high temporal resolution approaches. The authors spent a large part of the paper discussing a frame-shift approach that was used with a 200Hz camera to more accurately identify the transition points between behaviors. But this section ended concluding “Given the excellent performance at sampling rates at or above 50fps - and the impetus to use less-specialized cameras - we implement 60fps throughout the remainder of the manuscript.” I left this section confused, is the point of the extensive description of the frame-shift to describe something that one might use? Or to show that it is not necessary? I presume the former but I’m not sure when one might use it, given that the authors don’t use it for the rest of the paper. This should be clarified.

3) The authors spent a long time discussing and optimizing approaches for more accurately identifying the start and stop (or transition points) of behavior. However, their analysis depended on the same video they were ultimately analyzing, so I found it to be circular. Ideally they would know the “ground truth” of when the behavior actually started, which would need to be collected with an orthogonal approach such as an EMG recording. I think they should perform such recordings, or greatly reduce the analysis and discussion of this point in the paper, as they don’t have a ground truth to know if their manipulations provide more accuracy, vs. more temporal resolution but potentially less accuracy. As an alternative to recording EMG, the authors could program some type of simple robot to behave like a mouse, thereby also obtaining ground truth timing data to compare their video analysis to.

4) I was unclear of how many mice contributed to the 35 recorded units in that analysis. The number was not given, and from the way it was worded it seemed like it may have been just one mouse? The authors should report how many mice were used and test if their conclusions are robust across multiple mice.

5) For the A2A-caspase experiment, the authors focused on grooming. However, these neurons are strongly implicated in locomotion and hyperactivity. A classic paper that ablated these neurons

demonstrated hyperactivity (Sano et al., DOI: 10.1523/JNEUROSCI.23-27-09078.2003). It was not clear why the authors focused solely on grooming in these mice. Did the authors examine any of the other behaviors they classified with B-SOiD? Did B-SOiD pick up on a hyperactivity phenotype in these animals?

6) I was unclear on whether the same clustering be applied to multiple videos recorded in the same configuration. For instance, does cluster #1 always identify the same behavior across different overhead videos? If the clustering is specific to each video, does a user need to inspect the clusters on each video to assign them to different behavioral categories? Are there any pitfalls with comparing a specific behavior “ie: grooming” between videos if they are extracted on different low-dimensional axes? This is an important point that should be clarified.

Minor points:

1) The description, “standard 128GB RAM CPU found on a Desktop” should be edited, as it is not “standard” in my opinion to have 128GB of RAM on a Desktop.

2) Some figures were not well documented. Ie: in figure 2B, what do “true labels” refer to? How was this determined? I could not find this axis label or what it meant in the captions, text, or supplement.

Reviewer #2 (Remarks to the Author):

The manuscript by Hsu and Yttri represents an important leap forward in supervised classification of behaviors during free spontaneous exploration in rodents. The entire field of behavioral neuroscience is clamoring for such tools, and B-SOiD might be an answer for many diverse research groups. The authors show the B-SOiD resolves distinct behaviors better than Motion Mapper, but perhaps this was expected as that platform was built to work with the fruit fly. The closest, most recent platform to B-SOiD is MoSeq. MoSeq is proprietary, and for those who are able to work with this platform, reports are that it is nearly impossible to use without major computational prowess. The field needs platforms that are accessible and easy to setup for biologists without in-house computer scientists on their teams. B-SOiD appears compatible with commonly used DNNs like DeepLabCut and the user interface and use of this platform seems rather straightforward. Although this space of platforms for measuring animal behavior is becoming quickly crowded, I do foresee wide application of B-SOiD. The authors should consider the comments below prior to publication.

Major concerns

1. Authors mentioned that there are instances when a body part is occluded and that B-SOiD recognizes this as a null signal, and can use this to compute 3D behavioral structure. I remain confused about how B-SOiD does this without making errors. Authors should add more clarification.
2. I am confused about N numbers and animal genotypes. Methods say 6 C57BL/6 mice were used in studies, yet Figure 2 shows a brown mouse. Which strain is this and how many mice used?
3. The single frame images in Figure 2 are not good representations of itch. Perhaps frames showing the animal using their hind limbs to scratch the body would clearly demonstrate itch behavior. Related to this, it was hard to detect the itch behavior in Video 1, although B-SOiD clearly labeled “Itch” in annotation. Automatic measurement of itch would be monumental for the pain/itch research community. Authors should consider injecting an itch-inducing compound in the mouse like histamine or chloroquine to show that B-SOiD accurately detects heightened itch as well as a human observer.
4. Investigate vs locomotion is separated out by B-SOiD, but analyzing Video1, it is hard for me to tell the difference between these two behaviors.
5. For the 11 distinct behaviors identified in Figure2, does B-SOiD produce data on frequency or time animals are engaged in such behaviors? It appears this is the case based on how data is plotted in Figure 6. It would be nice to have such data associated with the 11 behaviors in figure 2 as well, as this would provide the most meaning to a biologist.

Minor concerns

1. Authors say several times that any model organism can be used, but only show rodents. They should either show additional model organisms or tone down this language without evidence for this claim.
2. Authors end the abstract and results talking about the potential use of B-SOiD to studying pain, OCD, and motor control. While this is a lofty goal, they should remove this level of specificity without evidence.

Reviewer #3 (Remarks to the Author):

In this paper the authors propose an unsupervised approach to classify the kinematics of animal body parts in video recordings into behavioral categories. The proposed approach assumes that the videos have been previously ran over a pose-estimation algorithm (the authors use DeepLabCut) to subsequently perform a dimensionality reduction of the space determined by the tracked body-parts into a low-dimensional space. Finally, the proposed methods uses a random-forest classifier to separate the dynamics of the low-dimensional features into a certain number of behavioral categories. For demonstration, the authors use their proposed method to classify mouse open field behavior from bottom-up video recordings in an empty cage with simultaneous electrophysiological recordings. The authors use a down-sampled frame shifting approach to increase the temporal precision of the behavioral classification of high-speed camera recorded videos (they also compared the results when using top-down videos). By shifting the down-sampled videos, it is possible to preserve some high-temporal resolution of the behavioral classification so that it is possible to align the behavior transitions to the ephys signal. Finally, the authors compare B-SOiD against MotionMapper, another unsupervised behavioral classification approach based on body parts kinematics (extracted using DeepLabCut).

Although unsupervised approaches of animal behavior classification are of great value in neuroscience and computational ethology research to reduce the experimenter bias and reduce their time-cost, the manuscript in its current form lacks clarity in general. Due to a poor organization of the text, the description of the methodology and results make the evaluation of its contribution difficult. I recommend that before considering this manuscript for publication, the authors need to improve the following concerns:

Major

The overall organization of the manuscript should be improved significantly. The authors do not provide a clear structure of the text. It is very difficult to follow the motivation and validity behind the ideas the authors are trying to convey.

A clear description of the rationale of the results should be presented. Perhaps by organizing the manuscript in a clearer way would help in this regard.

The methodology is very poorly described and organized. A clear sequential of the processing pipeline needs to be provided. A diagram should help significantly.

A better explanation about the purpose of pairing behavioral classification with electrophysiological recordings is needed. Although, this type of analysis is valuable for the study of the relationship between brain activity and behavior, this type of justification needs to be improved.

Methods:

A better organization of the Methods section is needed. Currently, the methods section starts with a description of the repository for the code instead of a general description which gives the reader a better sense of the whole procedure and helps to follow the organization of the subsections.

- A general diagram of the processing pipeline would greatly improve the description of the methodology.

- The description of the repository should be moved to the code availability section at the end of the Methods section.

- The code is hosted in two different repositories. It would be nicer if it was kept in only one.

Data processing feature extraction.

A better introduction to this subsection is needed. Currently, there is only pseudo-code to describe a couple of algorithms without explaining why these algorithms are used in the first place.

A description of all variables in the algorithm description is needed (e.g. what are L, D, p, etc)
The fact that pose-estimation is needed before the methodology in this paper can be applied (e.g. DeepLabCut, LEAP) needs to be explicitly clarified further. Although references are included, such methodologies should at least be cited here and briefly explained.
The authors mention sampling frequency of 60 Hz but they haven't explained the parameters that the description of the data processing are referring to. Are they already referring to a particular dataset? All these specs should be clarified in advance.
A better overall explanation of all the parameters referred to in the text needs to be provided (e.g. what 10 Hz window? What fragments?).

Dimensionality reduction

Again, since there is no big picture description of the methodology, it is difficult to understand the justification of this procedure (e.g. why is required to reduce the dimensionality?) where in the processing pipeline is the dimensionality reduction taking place?
The authors seem to start a UI description (e.g. start clustering label) that was never mentioned or explained before. If a description of the methods will be based on the UI and its functionality, then a previous introduction and description of the UI is needed.
In general, the authors used external algorithms that need to be properly described and references should be cited (e.g. UMAP, HDBSCAN, Random forest classifier).
Behavioral subjects and experimental set-up

The experimental setup description should be explained before the algorithm description so that the readers know what data has been used and how it was collected. The subsections of the Methods should be better organized.

Electrophysiology

The authors justify including ephys data to "demonstrate differences in sample resolution". This is not clear. A better explanation is needed (e.g. what sample resolution? what does ephys add to video recordings for behavioral classification?)

Motion energy image mean-squared-error

A better description of the reason behind calculating this measure and of the definition is needed.

Results

The organization of the section should be greatly improved. This section starts with a very vague description of B-SOiD followed by a description of supplementary figures. There is no introduction to the section or a general map to guide the reader about the experiments and results carried out. A better description of the algorithms used needs to be provided also here (given that the Methods section is not well organized). For example, what are UMAP and HDBSCAN and why were they chosen? How much does this selection change the overall results?

A better description of this part in the Methods would help so that it can be referred to in the Results (e.g. extract what are separate regions in the behavioral space? does this happen on the reduced manifold?, etc)

The authors first seem to use UMAP to reduce the dimensionality of the pose-estimation features to classify them into different behaviors but then argue that doing this introduces problems that they propose to solve doing the classification over the high-dimensional space. This is very confusing and needs to be clarified. Again, the poor description in the methods sections makes it difficult to follow the sequence of the results. Similarly, the authors argue that to improve accessibility, they now present a downloadable app never mentioned before.

B-SOiD extracts behavioral clusters in high-dimensional space

This subsection describes more the methodology than the methods section. This should be moved to the corresponding section and referred to it in the Results.

Discussion

A better organization of this section is needed. Currently, there is no guide for the author about the topics being discussed. Sometimes the text reads more like part of the Methods. Perhaps subtitles could help to improve the organization.

Minor comment

Include line numbers so that suggestions on how to improve the text can be more specific.

Reviewer #1 (Remarks to the Author):

This manuscript by Hsu and Yttri describes a new approach for quantifying animal kinematics, clustering pose information to identify specific behaviors and how they change across time. While there have been huge strides in pose estimation from video in recent years, the next level remains a challenge: what should we do with the pose data? How can it inform us about how the animals are behaving across time?

We thank the reviewer for the following thoughtful comments and agree with the pressing need to answer *What to do with all this position data?*.

1) The authors claim to demonstrate B-SOiD's use "in a variety of experimental models (mouse open field behavior, rat reach to grasp task, and human kinesiology data)". However, for two of these applications (rat reach to grasp task and human kinesiology data) they show almost no data, only a supplemental figure or video without description of what was being shown or concluded from these. It was a bit easier to see in the human example that they were identifying different poses but I was confused about the take home point of the rat reaching figure, or how to interpret the figure as this was not well described in the text. Was the inclusion of these just to demonstrate that they can run B-SOiD on poses extracted from videos other than mouse open-field? Did they learn anything about rat reaching or human exercise through these analyses? As it stands, I don't think it's fair to claim that the use of B-SOiD has been demonstrated in rats and humans, as the examples they show are anecdotal.

The human movement and rat reaching examples were intended as proof of concepts to demonstrate feasibility beyond a mouse in the open field. The ability to segment behaviors actually should be of little surprise, as B-SOiD only knows of the spatiotemporal patterns of points – not 'left foot', 'locomotion', or even 'mouse'. Additionally, they showcase the usability of other pose estimation packages beyond DeepLabCut, specifically OpenPose, and now LEAP. We have expanded the explanatory text but more importantly, we've taken this comment as an opportunity to expand the depth of the examples, specifically:

- The rat reach data previously demonstrated identified sub-actions, and their consistent patterning throughout dozens of reaches, as well as a plot of the spatiotemporal location of sub-actions during an example successful and failed reach attempt. We have added a directed graph that identifies how the pattern of transition from one sub-action to the next changes throughout the course of learning. These quantifications reveal that the behavioral structure we previously identified emerges over time with practice, rather than being inherent to the action itself.

- The human data now includes Sup Video 3, allowing the comparison across individuals and the behavioral performance to classification. The consistency of the model across individuals can be easily seen in the correct action classification across subjects as well as the conserved location of these points in UMAP space. Additionally, this data demonstrates that B-SOiD can work with OpenPose data, rather than only DeepLabCut.

- We have also added B-SOiD segmentation of fly behavior. Like the human data, we demonstrate consistent classifications across individuals while also adding to the pose-estimation inputs the code can work with (LEAP).

As this is a methodological paper, we do not claim that these examples great depth in understanding behavior. Instead, the tool enables future studies to do so, and we provide here a qualitative quantification of B-SOiD implementation across sessions - rather than an anecdotal N of 1. In the case of the rat, a thorough interrogation is underway in collaboration with the Leventhal group which provided the data, some of which can be seen in the evolution of learning structure shown here. While tempting, we decided against demonstrating novelty experiments - such as demonstrating that the senior PI performs squats more slowly – as this would cheapen the science and detract from the main point that B-SOiD can deliver these critical data. Despite this limitation, we find that the results are quite striking and even in their limited capacity would be difficult to impossible to achieve by any other means. We hope the reviewer agrees that these additions surpass the bar to add credence to our claim.

2) There appeared to be a tension in the paper between wanting to provide a simple tool that could be used cheaply and easily by many people with off the shelf cameras, vs. a high resolution tool that could be used for synchronizing pose information with electrophysiology or other high temporal resolution approaches. The authors spent a large part of the paper discussing a frame-shift approach that was used with a 200Hz camera to more accurately identify the transition points between behaviors. But this section ended concluding "Given the excellent performance at sampling rates at or above 50fps - and the impetus to use less-specialized cameras - we implement 60fps throughout the remainder of the manuscript." I left this section confused, is the point of the extensive description of the frame-shift to describe something that one might use? Or to show that it is not necessary? I presume the former but I'm not sure when one might use it, given that the authors don't use it for the rest of the paper. This should be clarified.

We thank the reviewer for recognizing this point. To main goal is as follows: frameshifting allows a means to robustly classify behavior at the native camera temporal resolution without succumbing to the low signal-to-noise encountered

with high framerates, e.g. the interframe differences are very low. Even if the framerate of the camera was 30fps, the frameshift method (reduce to 10fps for robustness, then shift a frame and average to get back up to native resolution) is beneficial. We have re-written the section in full to help define our purpose and the rationale of using 60fps. We include a particularly relevant portion of that text here:

“We quantified the effect of different temporal resolutions by first subsampling a 200fps video, thus providing an internally consistent comparison across resolutions (e.g. to extract 20fps for analysis, we used every tenth frame of the 200fps video). B-SOiD was then run independently on each resolution of the video, using frameshifting on each version. With the original 200fps as a standard, we observed that the predictions across resolutions was highly coherent. Even at 10fps (which was not further downsampled/frameshifted), we observed a median coherence of 84% across behavioral groups (Fig. 2f). As sampling rate increased, coherence improved - although the added benefit of increased sampling rate plateaus after 50fps. These changes can be attributed to a reduction in start/stop time inaccuracies, as seen in Fig. 2g and Supplemental Fig S3. The frameshift paradigm allows B-SOiD to predict behaviors at a temporal resolution matching the sampling rate of the original video, enabling a notably deeper analysis of action kinematics (Fig. 2d, g). Given the excellent performance above 50fps and the impetus to use less-specialized cameras, we the remainder of this manuscript focuses on easily attainable 60fps video..”

3) The authors spent a long time discussing and optimizing approaches for more accurately identifying the start and stop (or transition points) of behavior. However, their analysis depended on the same video they were ultimately analyzing, so I found it to be circular. Ideally they would know the “ground truth” of when the behavior actually started, which would need to be collected with an orthogonal approach such as an EMG recording. I think they should perform such recordings, or greatly reduce the analysis and discussion of this point in the paper, as they don't have a ground truth to know if their manipulations provide more accuracy, vs. more temporal resolution but potentially less accuracy. As an alternative to recording EMG, the authors could program some type of simple robot to behave like a mouse, thereby also obtaining ground truth timing data to compare their video analysis to.

The reviewer brings up an instructive point, though we suggest here a compromise solution. EMG would provide ground truth for start-time of a single movement, e.g. the loading of muscles that leads to the initiation of a limb movement. We do note that this signal does not readily distinguish between behaviors – e.g. head vs face grooming, turning vs walking. These are much better, or potentially only distinguishable by body angle and relative limb distances. This problem becomes more poignant at transitions, where a limb may be moving in both contexts. To identify particular muscle activation patterns, something akin to a ‘EMG B-SOiD’ would likely be necessary. As such, it is unclear how EMG would provide a useable ground truth except for select examples, e.g. a transition from stationary to moving.

If we are to accept this limitation, focusing only on instances when clear changes in muscle activation occur, EMG becomes equivalent to a limb speed threshold. In this case, the frame-to-frame displacement (aka speed) delivers the same signal. We note that this frame-to-frame limb speed signal is different than what B-SOiD uses. B-SOiD uses a 100ms composite of three parameters: speed, inter-point angle, and inter-point distance.

Thus, to address the reviewer's important point, we aligned the frame-to-frame limb displacement of all detected actions using either 10fps or the frameshifted 60fps – similar to the to the individual transition examples in Fig 2d,g. Importantly, the resolution is identical (60fps), only the alignment granularity differs. If alignment accuracy increases, we would expect sharper changes in speed that are not smeared over time. These changes should also occur at the specific moment of transition. This is exactly what we found, and include this output in Supp Figure S4. For ease, we include the most intuitive example here – locomotion - focusing in on the critical period over which frameshifting should improve transition time accuracy, 100ms before and after alignment. Right limbs are orange, forelimbs are darker than hindlimbs, high resolution is on the right. Note the sharp inflection of the darker forelimb traces at t=0 on the right, and the overly smooth, slightly misaligned traces on the right.

If instead the goal of using EMG was to determine when the animal first shows the intention of movement, we would suggest that the brain is preferable to muscle. The clearly aligned M1 neural activity modulations provide a valuable movement-related signal. The presence of well-aligned neural signatures preceding movement, and that signal-to-noise is better at 60fps than 10fps, strongly indicates improved temporal accuracy with increased resolution. Moreover, it indicates that the B-SOiD-identified groups are neurophysiologically meaningful, something that needn't

have been true - discussed in further detail in the response to the next comment. Finally, the exceedingly clear patterns in motion energy patterns seen in Fig 5A provides additional evidence for excellent alignment accuracy. We acknowledge that there is no comparison of framerates here, only examples of alignment accuracy. Even with the data taken from the first 300-600ms of movement, most readers will recognize the behavior in each panel without needing to read the group ID. If temporal (or behavioral) accuracy was inadequate, we should see images muddled by overlapping with extraneous behaviors, not clear patterns.

Finally, we have attempted to adjust our text so as to reflect the reviewer's comments

4) I was unclear of how many mice contributed to the 35 recorded units in that analysis. The number was not given, and from the way it was worded it seemed like it may have been just one mouse? The authors should report how many mice were used and test if their conclusions are robust across multiple mice.

Thank you for catching this omission. A single session was used, and additional details have been added to the Results and Methods. In case the reviewer is interested as to why only one session was used, the purpose was to have the the same actions contributing to the neural responses across the same units, thus eliminating any obstacles to interpreting the neural representation and functional utility of improved temporal alignment.

We note here, and have clarified in the text, that our objective was not to derive a particular conclusion about the function of M1. As the reviewer rightly states, doing so would certainly be more robust if demonstrated across multiple mice and is beyond the scope of this paper. This is further recognized in text added to the current manuscript, and is a goal for a neural correlates paper we hope to submit by year's end.

Our first objective was to demonstrate that our classifications and alignments are supported by physiological signals. From the previous text:

"In this first demonstration of motor cortical activity aligned to the breadth of naturalistic behaviors observed, we noted distinct neural signatures for the range of identified naturalistic behavior groups".

We fully recognize that just because behavioral clusters are clearly statistically distinct, there are no assurances that the brain cares about these distinctions (*"More importantly, the clear population responses support that the mathematically-established B-SOiD groupings reflect real distinctions at the level of neural representation"*). Moreover, the presence of robust neural responses support our claims of behavioral accuracy. To this end, we have supplemented these results by providing PETH's from the same neural data - but with randomized alignments (Sup Fig 4). This approach provides a baseline from which to determine if the B-SOiD generated alignments represent real neural phenomena. When randomized, the strong temporal patterns are completely lost, manifesting an average peak magnitude an order of magnitude smaller. These data support our conclusion that the start times, and the mathematically derived classifications themselves, represent real neurophysiological states. While additional sessions may increase the robustness, we find these simple and robust claims are well-founded, particularly given the newly provided analyses.

The second objective was to demonstrate the 'real-world' benefit of increased temporal resolution for alignment of neural activity. Specifically, neural signal to noise should increase if the improvements in temporal resolution depicted in Figure 2 provide a real advantage. We note a marked improvement in neural activity for alignments derived with 60fps resolution compared to 10fps. In the modeled neural data, much like the empirical behavioral data of Fig 2f, 60fps yields near ground-truth results. Given that these conclusions are tested via within-session analyses, not likely to be different in another animal, and that the N of the alignments under consideration are in the thousands (albeit from a single animal), we decided to present a single dataset.

5) For the A2A-caspase experiment, the authors focused on grooming. However, these neurons are strongly implicated in locomotion and hyperactivity. A classic paper that ablated these neurons demonstrated hyperactivity (Sano et al., DOI: 10.1523/JNEUROSCI.23-27-09078.2003). It was not clear why the authors focused solely on grooming in these mice. Did the authors examine any of the other behaviors they classified with B-SOiD? Did B-SOiD pick up on a hyperactivity phenotype in these animals?

The requested locomotor data did appear in the original version, although placed in Sup Figure S6 and without mentioning the direct effects.

"We were also able to uncover other kinematic effects, including for itching and locomotion behaviors(Fig. S6 – now Figure S8)."

We now include a discussion of those results. We confirm the locomotor hyperactivity, and demonstrate that these effects are due to increased stride length but not stride speed, in additional text:

“ We were also able to uncover other kinematic effects (Fig. S6), including pronounced increases in itching speed. Locomotor stride length, but not stride speed was also significantly increased, providing a kinematic mechanism for previous seminal motor control work that observed gross locomotor hyperactivity following indirect pathway lesion (Sano et al).”

We focus on grooming in the main manuscript in order to highlight the specific benefits of B-SOiD, demonstrating that B-SOiD provides hitherto difficult or impossible measures of grooming:

“Current quantitative methods for grooming only provide bout duration and typically the canonical grooming types are combined because of technological limitations. Realizing these robust but hitherto indiscernible effects is made possible because of B-SOiD’s ability to dissociate groom types, measure kinematics, and accurately identify the start and stop of bouts.”

6) I was unclear on whether the same clustering be applied to multiple videos recorded in the same configuration. For instance, does cluster #1 always identify the same behavior across different overhead videos? If the clustering is specific to each video, does a user need to inspect the clusters on each video to assign them to different behavioral categories? Are there any pitfalls with comparing a specific behavior “ie: grooming” between videos if they are extracted on different low-dimensional axes? This is an important point that should be clarified.

Yes, one of the fundamental benefits of B-SOiD is that once the B-SOiD model is trained, it can be used on any videos with that orientation (See right side of Fig 1a, “Apply classifier to new data”). Cluster 1 will always be the same output for that model.

We have clarified this point in the introduction:

“More importantly, once trained, the algorithm can generalize across animals, cameras, and setups, thus solving the issue of transference.”

And this to Figure 1’s legend

“Once trained, the model will segment any video into the same groupings.”

Additionally, we have added a process pipeline diagram, we hope this is made more clear.

Minor points:

1) The description, “standard 128GB RAM CPU found on a Desktop” should be edited, as it is not “standard” in my opinion to have 128GB of RAM on a Desktop.

We agree, and ‘Standard’ has been removed.

2) Some figures were not well documented. Ie: in figure 2B, what do “true labels” refer to? How was this determined? I could not find this axis label or what it meant in the captions, text, or supplement.

We apologize for this convention in annotating training data as true label and testing data as predicted labels, and realize that it may not be clear how the designation is derived. We have now clarified in the text:

“To test whether these pose relationships can be learned accurately, we tested the mapping on randomly selected 20% of the data. The predicted labels generated by our random forest classifier matched cluster assignments by HDBSCAN (‘true labels’) over 90% of the time.”

Reviewer #2 (Remarks to the Author):

The manuscript by Hsu and Yttri represents an important leap forward in supervised classification of behaviors during free spontaneous exploration in rodents. The entire field of behavioral neuroscience is clamoring for such tools, and B-SOiD might be an answer for many diverse research groups. The authors show the B-SOiD resolves distinct behaviors better than Motion Mapper, but perhaps this was expected as that platform was built to work with the fruit fly. The closest, most recent platform to B-SOiD is MoSeq. MoSeq is proprietary, and for those who are able to work with this platform, reports are that it is nearly impossible to use without major computational prowess. The field needs platforms that are accessible and easy to setup for biologists without in-house computer scientists on their teams. B-SOiD appears compatible with commonly used DNNs like DeepLabCut and the user interface and use of this platform seems rather straightforward. Although this space of platforms for measuring animal behavior is becoming quickly

crowded, I do foresee wide application of B-SOiD. The authors should consider the comments below prior to publication.

Excellently put, thank you.

Major concerns

1. Authors mentioned that there are instances when a body part is occluded and that B-SOiD recognizes this as a null signal, and can use this to compute 3D behavioral structure. I remain confused about how B-SOiD does this without making errors. Authors should add more clarification.

We thank the reviewer for recognizing this missing clarity, and have taken the opportunity to expand the discussion of this point in the methods. In doing so, we hope to have clarified the approach and showcased the utility of machine learning.

“ Integration of low-confidence pose estimates

The occlusion of a point from view can be informative (e.g. during a rear, the snout is often occluded by the body when viewed from below), but a missing point can be the result of poor pose estimation on a given frame. The certainty of each frame's pose estimation is provided by most pose estimation software in the form of prediction confidence/likelihood values. In all data sets, we observed a bimodal distribution comprised of either very high or low confidence values. To bisect the two distributions with each session, B-SOiD designates all points with a confidence score below the elbow point (difference (high - low) between adjacent likelihood becomes positive) of the probability. We remove these low-confidence points and substitute that position with last high-confidence position. Thus, the displacement between frames for a low-confidence point is zero. It should be noted that even when stationary, pose estimation programs do not output identical positions in consecutive frames.

Low confidence estimations can occur from a missing body part (e.g. from being behind another body part) or poor prediction (e.g. blurry video or inconsistent lighting). In the latter case, the aberrations which led to the low-confidence points are typically short lived, often only a single frame, and are fully mitigated by averaging over the 100ms frameshifting interval. Prolonged low-confidence points contribute a spatiotemporal signature, and if repeated as a behavior, may be part of an identified spatiotemporal pattern (either in training or prediction). The occlusion of the snout during rear(+) is one example. Importantly, occasional prolonged aberrations do not adversely affect the algorithm. During training, spurious omissions will be too variable to constitute a conserved pattern - and the added variability may make the random forest even more robust. During prediction, the trained model utilizes 36 spatiotemporal features, minimizing the effect of a pose estimation error. This ability to incorporate patterned omissions and overcome spurious errors is another benefit of adding the trained classifier following the clustering.”

2. I am confused about N numbers and animal genotypes. Methods say 6 C57BL/6 mice were used in studies, yet Figure 2 shows a brown mouse. Which strain is this and how many mice used?

We apologize for this oversight and have fully characterized the mice. Although the vast majority of C57BL/6 mice are quite dark in color, there exist substrains within C57BL/6; the 6N substrain specifically is known to produce brown animals (see Fontaine and Davis, *Diabetes* 2016). Although the sub-strain information is not available for this animal (nor a vast majority of studies), we have stated “*The brown mouse used in Figure 2a is an example of the diversity of the C57BL/6 line, specifically the substrain 6N (NIH) lineage, which can produce brown fur.*” in the Methods.

While these differences are important in the study of some topics (JAX provides an excellent summary on their website) we do not believe these differences detract from our findings. Rather, the excellent generalizability of B-SOiD is showcased; the brown mouse's data is part of the accuracy benchmarking in 2b,c. This point is now emphasized in the text.

3. The single frame images in Figure 2 are not good representations of itch. Perhaps frames showing the animal using their hind limbs to scratch the body would clearly demonstrate itch behavior. Related to this, it was hard to detect the itch behavior in Video 1, although B-SOiD clearly labeled “Itch” in annotation. Automatic measurement of itch would be monumental for the pain/itch research community. Authors should consider injecting an itch-inducing compound in the mouse like histamine or chloroquine to show that B-SOiD accurately detects heightened itch as well as a human observer.

We have confirmed the quality of the ‘itch’ category with two individuals we consider to be experts noted pain researchers Dr. Sarah Ross, who we have added to the acknowledgements. She confirms the B-SOiD-derived group as being appropriate for “itch”, and that the videos provided in the paper's link are excellent. She did agree with the reviewer in that the brown mouse example in 2a was sub-optimal, particularly because most researchers will not be familiar with viewing this behavior from below. Rather than attempting to capture the variety of body poses underlying

the behavior, we have substituted a more canonical image with the hindpaw at the nape of the neck. The black mouse, she thought, showed the hindfoot moving towards the head, indicative of itch.

We have also provided a written summary of this and every behavior in the figure legend to Sup Video 1. Itch is summarized as: “*distance between one hindpaw and snout greatly reduced while the displacement of this same paw is greatly increased.*”

4. Investigate vs locomotion is separated out by B-SOiD, but analyzing Video1, it is hard for me to tell the difference between these two behaviors.

Realizing that the many histograms of feature distributions in the link provided may be daunting, we have provided a brief summary of each category extracted by B-SOiD in the figure legend to Sup Video 1. In the case here, a prominent distinction is that the hind paws do not move while head extends; and unlike locomotion, the movement is more of an extension of the upper body/neck rather than translation of the body. These included descriptions may also help with the previous point. Additionally, the link includes videos of randomly chosen example behaviors, though we acknowledge that realizing the difference in a few individual videos without prior knowledge may not be obvious.

5. For the 11 distinct behaviors identified in Figure2, does B-SOiD produce data on frequency or time animals are engaged in such behaviors? It appears this is the case based on how data is plotted in Figure 6. It would be nice to have such data associated with the 11 behaviors in figure 2 as well, as this would provide the most meaning to a biologist.

Thank you for this excellent idea. We have added to Figure 1c the average duration of each of the behaviors throughout that hour-long dataset.

Minor concerns

1. Authors say several times that any model organism can be used, but only show rodents. They should either show additional model organisms or tone down this language without evidence for this claim.

We note that human data (Figure and video) appears in the supplement, as mentioned in the Results. This data uses OpenPose to provide pose estimation. Additionally, to help assuage the reviewer, have added fly data from the Mala Murthy and Josh Shaevitz labs, also in the supplement.

2. Authors end the abstract and results talking about the potential use of B-SOiD to studying pain, OCD, and motor control. While this is a lofty goal, they should remove this level of specificity without evidence.

We appreciate this comment, but we contend that this manuscript/algorithm provides “behavioral and kinematic measures [that] are difficult but critical to obtain, particularly in the study of pain, OCD, and movement disorders.”

Note we are referring to the value of the behavioral and kinematic values, e.g. the presence of a head groom, and the peak speed of each stroke of that groom. The reviewer previously noted that “Automatic measurement of itch would be monumental for the pain/itch research community.” We agree, and demonstrate here a) automatic detection of itch and b) detailed kinematic parameters concerning the peak speed and distance of itching - not across an itch bout, but of every stroke comprising that bout (Sup Fig S6). The same can be said about grooming, a critical metric to the OCD research community - and featured in Fig 6 for exactly this reason. Several prominent groups studying the circuits underlying OCD, including Drs. Susanne Ahmari and Ann Graybiel, are currently using B-SOiD. For similar reasons, the “kinematic measures of individual limb trajectories” in question are of use to study of movement disorders. We are currently using B-SOiD to perform in-depth quantification of a Parkinsonian mouse and potential therapeutics. The A2A-lesion mice were created by Dr. Aryn Gittis in her attempts to understand movement disorders, the driving force of her lab. Finally, although the rat reach-and-grasp data shown in the supplement are only control animals, Dr. Dan Leventhal is a neurologist and we are working with his lab on a studying comparing these data with Parkinsonian rats. We respectfully maintain that B-SOiD provides kinematic and behavioral **measures** that are critical and in dire need to these lines of studies. If the reviewer is still in disagreement, would “potentially critical” suffice?

Reviewer #3 (Remarks to the Author):

The overall organization of the manuscript should be improved significantly. The authors do not provide a clear structure of the text. It is very difficult to follow the motivation and validity behind the ideas the authors are trying to convey.

We apologize that the reviewer feels this way and hope the following changes assist in the reading. Your comments provided an instructive opportunity to improve our clarity and purpose. Our motivation is, as the reviewer states, to ‘propose an unsupervised approach to classify the kinematics of animal body parts in video recordings into behavioral categories’ that is ‘of great value in neuroscience and computational ethology research to reduce the experimenter bias and reduce their time-cost’. This message can be found in the abstract (“*To provide the missing bridge from*

poses to actions and their kinematics, we developed B-SOiD – an open-source, unsupervised algorithm that identifies behavior without user bias.”).

We hope to have provided additional clarity through the new beginning to the results, reformatted discussion, and the inclusion of a processing pipeline diagram.

The methodology is very poorly described and organized. A clear sequential of the processing pipeline needs to be provided. A diagram should help significantly.

We thank the reviewer for the excellent idea. A detailed diagram of the processing pipeline now appears in the supplement.

A better explanation about the purpose of pairing behavioral classification with electrophysiological recordings is needed. Although, this type of analysis is valuable for the study of the relationship between brain activity and behavior, this type of justification needs to be improved.

We have worked to make this justification more clear. Specifically, we have added the following text and figures:

“Improved temporal resolution is a critical advancement for analyzing neural correlates of spontaneous behaviors. To assess the real-world benefit of increased temporal resolution, we simultaneously recorded 35 units....”

Additionally, the recordings provide critical evidence that the mathematically derived B-SOiD classifications represent real neurophysiological states – a fact that should not be assumed. We hope that the next makes this justification more clear.

Methods:

A better organization of the Methods section is needed. Currently, the methods section starts with a description of the repository for the code instead of a general description which gives the reader a better sense of the whole procedure and helps to follow the organization of the subsections.

We have given a general description to start the methods and started in, followed by a focus first on the animal, not code, as requested.

- A general diagram of the processing pipeline would greatly improve the description of the methodology.

This has been added as Supp. Figure S1. Thank you for the recommendation.

- The description of the repository should be moved to the code availability section at the end of the Methods section. This has been moved.

- The code is hosted in two different repositories. It would be nicer if it was kept in only one.

Data processing feature extraction.

A better introduction to this subsection is needed. Currently, there is only pseudo-code to describe a couple of algorithms without explaining why these algorithms are used in the first place.

A description of all variables in the algorithm description is need (e.g what are L, D, p, etc)

We apologize for not making the definition of these variables easier to find, originally stated just after the pseudocode:

“We then computed all pair-wise distances from 6 points (15 pairs, L), angular change within each pair over time (15 deltas, Θ), and individual body part displacement over time (6 deltas, D), as described in Algorithm 1. “

We now clearly state their values at the beginning of this section in the methods. Again, the reviewer’s recommendation to include a process pipeline should help as well, and these values have been identified in that plot. p , found in the discussion of likelihood distributions, is the common term to refer to probability (high likelihood = probability of 1). We acknowledge this ambiguity and have replaced this section with clearer language to avoid confusion.

The fact that pose-estimation is needed before the methodology in this paper can be applied (e.g. DeepLabCut, LEAP) needs to be explicitly clarified further. Although references are included, such methodologies should at least be cited here and briefly explained.

The following sentence with citations has been added to the introduction, where pose estimation software is introduced:

“Recent advances in computer vision and machine learning have enabled automatic tracking of body part positions.”

Additional detail has been added throughout the text.

The authors mention sampling frequency of 60 Hz but they haven't explained the parameters that the description of the data processing are referring to. Are they already referring to a particular dataset? All these specs should be clarified in advance.

A better overall explanation of all the parameters referred to in the text needs to be provided (e.g. what 10 Hz window? What fragments?).

We apologize that the language used here was not clear. 'Fragments' was intended to describe the divisions of behaviors captured between consecutive frames. There is no particular dataset to note. We have now clarified that portion of the text and removed the term 'fragments'. 10Hz has been replaced by 10fps so as to make clear that this refers to the downsampling discussed thoroughly throughout the Results and Methods.

We have also re-written this section in the results, and hope that you find that the following clarifies the details:

“Accurate resolution of the timing of behavior transitions is a necessary feature of segmentation beyond identification. We present two example transitions (Fig. 2d), at 10 frames per second (fps) a temporal resolutions on par with many current methods. Although the group identification is correct, the large inter-frame interval misses the transition time, leading to much of the behavior being inaccurately categorized.

Resolving transitions with adequate precision for use with electrophysiological measures requires considerably faster sampling rates, which are unavailable given current technology. However, a particular challenge in defining behaviors at a high sampling rate is that pose location jitter dominates the signal from any movement (Fig. 2e (left)). It is precisely this loss of frame-to frame difference at high sampling rates that makes 10fps sampling a popular temporal resolution.

To resolve behavioral transitions at the under-ten millisecond-scale, we introduced a "frameshift" manipulation, borrowed from recent automatic speech recognition innovations (Bartkova2015ImpactStudies), (Fig. 2e (right)). Briefly, B-SOiD initially downsamples all video, regardless of framerate, to 10fps to achieve a high signal to noise ratio in the spatiotemporal dynamics of the markers. The process is then repeated, with each new set of predictions made on downsampled data, each time offset by an additional frame (t1, t2, t3 in Fig. 2e (right)). In essence, we decompose the high-resolution signal and run a sliding threshold for fitting the high-SNR decomposition. By combining behavior assignments extracted from the shifted, downsampled data, we gain improved transition time resolution while overcoming the hurdle of decreased signal-to-noise (Fig. 2f,g). Note, improving transition resolution does not fundamentally change the distribution of action durations observed at 10fps. Thus frameshifting carries over the robust behavioral signal provided by lower sampling to the native resolution of the camera used.”

And continue by defining where and why we use 60fps video:

“We note that the additional benefit of increasing sampling rate plateaus after 50fps. Given the excellent performance above this rate and the impetus to use less-specialized cameras, we will proceed for the remainder of the manuscript with relatively common 60fps video.”

Dimensionality reduction

Again, since there is no big picture description of the methodology, it is difficult to understand the justification of this procedure (e.g. why is required to reduce the dimensionality?) where in the processing pipeline is the dimensionality reduction taking place?

The authors seem to start a UI description (e.g. start clustering label) that was never mentioned or explained before. If a description of the methods will be based on the UI and its functionality, then a previous introduction and description of the UI is needed. In general, the authors used external algorithms that need to be properly described and references should be cited (e.g. UMAP, HDBSCAN, Random forest classifier). A better description of the algorithms used needs to be provided also here (given that the Methods section is not well organized). For example, what are UMAP and HDBSCAN and why were they chosen? How much does this selection change the overall results?

We apologize for not understanding what is missing. UMAP's original reference(31) is cited in the Results, Discussion, and Methods, as well as a review paper describing how its selection changes clustering results. We also described it and state explicitly why UMAP was chosen:

- A) *“This [UMAP's] non-linear dimensionality reduction approach provides an improved ability to delineate high-dimensional data in low-dimensional space over linear methods^{31,36,37}. In particular, UMAP is preferred over t-SNE (t-Distributed Stochastic Neighbor Embedding) for its ability to preserve global pairwise distances in embedding. This feature is critical for users to manipulate behavioral delineation.”*
- B) *“Recently, Uniform Manifold Approximation and Projection for Dimension Reduction (UMAP) has proven to be more computationally effective and faster for reduction algorithm than t-SNE³¹. In addition to local*

structure preservation like t-SNE, UMAP preserves the "global structure", or long-distance embedding placement of the dimensionally reduced."

- C) *"In simpler terms, similar mouse multi-joint trajectory will retain its similarity visualized in the low-dimensional space. B-SOiD achieves this through UMAP, a state-of-the-art algorithm that utilizes Riemannian geometry to represent real world data with the underlying assumptions of the algebraic topology³¹"*

HDBSCAN is similarly cited (32) and explained. Additionally, a majority of the Discussion was devoted to discussing the advantages and appropriateness of UMAP, HDBSCAN, and their combination.

A) *"To segregate behavioral assignments in the 11-dimensional UMAP space, we employed a hierarchical clustering method - Hierarchical Density Based Spatial Clustering of Applications (HDBSCAN)³². Similar density-based clustering methods have been employed for unsupervised segmentation of behaviors in both vertebrates and invertebrates^{15,16,38–42}. However, HDBSCAN is particularly well-suited to address the inevitable variability in pose estimations, even with state-of-the-art software (see Methods for specific HDBSCAN parameters), enabling B-SOiD to purify the training data to assign every frame."*

- B) *"...HDBSCAN algorithm³². It is particularly useful for UMAP outlier detections as it recognizes subthreshold densities."*

The justification and use of a random forest is also described:

"To improve consistency, speed, and applicability in classifying behaviors, we equipped B-SOiD with a random forest classifier. The random forest classifier is well-suited for high-dimensional feature training and has been shown to predict low-dimensional representation of high-dimensional features well, particularly compared to potential alternatives like MLP or SVM³⁶(see Methods for classifier design)"

The citation here provides an excellent manuscript concerning describing the benefits of a random forest with UMAP. We also contend that random forest is a basic tool of the community, but will happily add a citation suggested by the reviewer.

We apologize again, but we hope to better understand what the reviewer finds lacking. We also hope that the new processing pipeline diagram, extended Methods and revised Results will add additional information and organization.

Behavioral subjects and experimental set-up

The experimental setup description should be explained before the algorithm description so that the readers know what data has been used and how it was collected. The subsections of the Methods should be better organized.

We now lead with experimental setup.

Electrophysiology

The authors justify including ephys data to "demonstrate differences in sample resolution". This is not clear. A better explanation is needed (e.g. what sample resolution? what does ephys add to video recordings for behavioral classification?)

Our first objective was to demonstrate that our classifications and alignments are supported by physiological signals. Moreover, the presence of robust neural responses support our claims of behavioral accuracy. To this end, we have supplemented these results by providing PETH's from the same neural data - but with randomized alignments (Sup Fig 4). This approach provides a baseline from which to determine if the B-SOiD generated alignments represent real neural phenomena. When randomized, the strong temporal patterns are completely lost, manifesting an average peak magnitude an order of magnitude smaller. These data support our conclusion that the start times, and the mathematically derived classifications themselves, represent real neurophysiological states. While additional sessions may increase the robustness, we find these simple and robust claims are well-founded, particularly given the newly provided analyses.

The second objective was to demonstrate the 'real-world' benefit of increased temporal resolution for alignment of neural activity. Specifically, neural signal to noise should increase if the improvements in temporal resolution depicted in Figure 2 provide a real advantage. We note a marked improvement in neural activity for alignments derived with 60fps resolution compared to 10fps. In the modeled neural data, much like the empirical behavioral data of Fig 2f, 60fps yields near ground-truth results

Sample resolution applies to assessing the benefit of 60fps over 10fps resolution. We have changed the term to "temporal resolution"

Motion energy image mean-squared-error

A better description of the reason behind calculating this measure and of the definition is needed.

We have provided an extend description, rationale, use and citation:

“The term “motion energy” that we had mentioned was introduced by Stringer et al., where “Motion energy, computed as the absolute value of the difference of consecutive frames” (Stringer, 2019). Since the animal is freely moving in the environment, starting pose alignment is necessary. Following image registration using estimated outline of animal at the start of each identified behavior, we compute the motion energy (ME, absolute value of the difference of all consecutive frames) using MATLAB command imshowpair, capped at 600ms for conciseness. We then performed weighted averaging for each bout to reconstruct a single ME image. In other words, each pixel in such reconstructed ME image represents the average absolute difference between consecutive frames at a given pixel location. Since there are multiple instances of each action, we want to see if such animal-centric average absolute difference is conserved between instances. To quantify consistency, we performed all pair-wise image mean-squared-error using MATLAB command immse. Essentially, the pixel difference between instances (ME images) will be coalesced into a single value (MSE). MSE is inversely proportional to consistency of animal movement for each identified action.”

Results

The organization of the section should be greatly improved. This section starts with a very vague description of B-SOiD followed by a description of supplementary figures. There is no introduction to the section or a general map to guide the reader about the experiments and results carried out.

We agree with the reviewer that start to the results section could be greatly improved. As such, we have modified the text accordingly. We now begin the section by laying out each phase of the results and their rationale.

“We provide here an open source tool to resolve distinct behaviors (Fig 1a). To achieve this end, we sought to make use of pose estimation software, which uses computer vision and machine learning to identify the location of body parts from video. These techniques have made huge strides in recent years, but making sense of those data remains difficult.

We begin with a summary of the behavioral classification/segmentation tool, its computational underpinnings, and basic benchmarking.

In addition to the extraction of behavior from poses, B-SOiD provides a signal processing method that provides temporal resolution matching the video frame rate. We demonstrate the utility of this increased resolution - enabling improved alignment that optimizes the resolution of the neural signatures of behavior. In doing so we also provide neurophysiological verification of the mathematical-defined groups.

We then quantify the algorithm's performance across different camera angles and compare it to the current state of the art. These measures also serve to validate the external and internal consistency of the method, respectively.

The manuscript concludes with a real-world example of B-SOiD's potential, providing automated reports of multiple canonical types of grooming and with kinematic readouts unavailable with the current state of the art.”

A better description of this part in the Methods would help so that it can be referred to in the Results (e.g. extract what are separate regions in the behavioral space? does this happen on the reduced manifold?, etc)

The authors first seems to use UMAP to reduce the dimensionality of the pose-estimation features to classify them into different behaviors but then argue that doing this introduces problems that they propose to solve doing the classification over the high-dimensional space. This is very confusing and needs to be clarified. Again, the poor description in the methods sections makes difficult to follow the sequence of the results. Similarly, the authors argue that to improve accessibility, they now present a downloadable app never mentioned before.

We previous mentioned the app in the first (now second paragraph) and devoted half of the Figure 1 to it. We are unclear where the app should be first mentioned and kindly ask the reviewer to indicate a better place so that we can make this change.

B-SOiD extracts behavioral clusters in high-dimensional space

This subsection describes more the methodology than the methods section. This should be moved to the corresponding section and referred to it in the Results.

Discussion

A better organization of this section is needed. Currently, there is no guide for the author about the topics being discussed. Sometimes the text reads more like part of the Methods. Perhaps subtitles could help to improve the organization.

We have revamped the discussion. Unfortunately, journal guidelines prohibit the use of subtitles.

Minor comment

Include line numbers so that suggestions on how to improve the text can be more specific.

This has been changed.

REVIEWER COMMENTS

Reviewer #1 (Remarks to the Author):

The authors were extremely responsive to my concerns and I have no further comments. Congratulations on a very nice paper, and thank you for providing a very useful tool to the neuroscience community!

Reviewer #2 (Remarks to the Author):

During this revision, the authors have thoroughly addressed my concerns and I support publication. I think this will be a very useful new tool for behavioral neuroscientists.